# Weight matrices compression based on PDB model in deep neural networks

Xiaoling Wu [* 1]   Junpeng Zhu [* 1]   Zeng Li [1]

## Abstract

Weight matrix compression has been demonstrated to effectively reduce overfitting and improve the generalization performance of deep neural networks. Compression is primarily achieved by filtering out noisy eigenvalues of the weight matrix. In this work, a novel **Population Double Bulk (PDB) model** is proposed to characterize the eigenvalue behavior of the weight matrix, which is more general than the existing Population Unit Bulk (PUB) model. Based on PDB model and Random Matrix Theory (RMT), we have discovered a new **PDBLS algorithm** for determining the boundary between noisy eigenvalues and information. A **PDB Noise-Filtering algorithm** is further introduced to reduce the rank of the weight matrix for compression. Experiments show that our PDB model fits the empirical distribution of eigenvalues of the weight matrix better than the PUB model, and our compressed weight matrices have lower rank at the same level of test accuracy. In some cases, our compression method can even improve generalization performance when labels contain noise. The code is avaliable at https://github.com/xlwu571/PDBLS.

## 1. Introduction

Deep Neural Networks (DNN) have achieved outstanding performance in many fields such as computer vision (Leek et al., 2022), speech recognition (Mohanty et al., 2022) and recommendation systems (Da'u & Salim, 2020). Deeper and wider DNN frameworks have demonstrated superior learning performance (Yang et al., 2019), but many of these networks are extremely over-parameterized and prone to overfitting. Recently, some low-rank compression techniques have been applied to avoid overfitting by removing small singular values of the weight matrix. (Xu et al., 2019)

---
[*]Equal contribution [1]Department of Statistics and Data Science, Southern University of Science and Technology, Shenzhen, China. Correspondence to: Zeng Li <liz9@sustech.edu.cn>.

*Proceedings of the 42$^{nd}$ International Conference on Machine Learning*, Vancouver, Canada. PMLR 267, 2025. Copyright 2025 by the author(s).

propose a training scheme to promote the filters' low rank by using nuclear regularization and achieves the low-rank approximation of the original filters. (Idelbayev & Carreira-Perpinán, 2020) emphasize determining the ideal rank for each layer and weight approximation via adding a penalty term of the rank. (Liebenwein et al., 2021) further propose a compression architecture using SVD for local layer compression and minimizing the maximum relative error across different layers for global compression. However, these methods rely on hyperparameters and are generally difficult to optimize.

Recently, to understand the training mechanism and generalization performance of neural networks, RMT has succeeded in providing a theoretical explanation and improving generalization ability by analyzing spectral properties of the weight matrices of neural networks, see (Berlyand et al., 2023; Martin & Mahoney, 2020). For weight matrix $\mathbf{W}_{n \times p}$, RMT is mainly employed to study the eigenvalues $\{\lambda_1 \geq \cdots \geq \lambda_p\}$ of $\mathbf{W}^T\mathbf{W}$, in particular the limit of $F_n^{\mathbf{W}^T\mathbf{W}}(x) = \frac{1}{p}\sum_{j=1}^{p}\mathbb{1}_{\{\lambda_j \leq x\}}$ called **Limiting Spectral Distribution (LSD)**. According to (Martin & Mahoney, 2021), the training process of weight matrices mainly goes through three phases: initial phase, bulk+spikes phase and heavy-tailed phase (Fig.1). The randomly initialized weight matrix $\mathbf{W}_0$ satisfies $\mathbb{E}\mathbf{W}_0^T\mathbf{W}_0 = \sigma_0^2\mathbf{I}_p$, where $\sigma_0^2$ depends on the initial distribution of the entries of $\mathbf{W}_0$. At this stage, the LSD of $\mathbf{W}_0^T\mathbf{W}_0$ matches Marčenko-Pastur (MP) law. During the training process, the information learned by the weight matrix is mainly reflected in a few spiked eigenvalues exceeding the MP edge $\lambda_+$, and the initial variance also changes from $\sigma_0^2$ to $\sigma^2$. In this phase, the PUB model which assumes $\mathbb{E}\mathbf{W}^T\mathbf{W} = \operatorname{diag}(\underbrace{\alpha_1, \ldots, \alpha_K}_{K \text{ spikes}}, \underbrace{\sigma^2, \ldots, \sigma^2}_{\text{bulk}}) :=$ $\mathbf{\Sigma}_{PUB}$, is often used to analyze the trained weight matrices. Correspondingly, $\mathbf{W}^T\mathbf{W}$ exhibits several comparatively large sample spiked eigenvalues, alongside clustered bulk eigenvalues. In the final training phase, the eigenvalues of $\mathbf{W}^T\mathbf{W}$ gradually follow a heavy-tailed distribution, but this phase is not very common. Thus, we mainly focus on the bulk+spikes phase.

Furthermore, (Staats et al., 2023) has verified that in the bulk+spikes phase, large spiked eigenvalues learn rules from data while small bulk eigenvalues do not contribute to model learning. They propose to compress network by removing

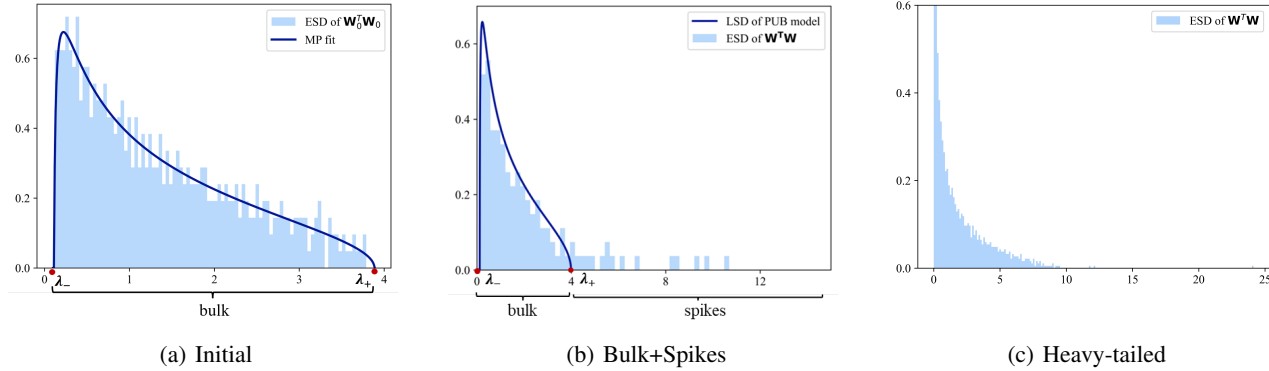

(a) Initial           (b) Bulk+Spikes           (c) Heavy-tailed

*Figure 1.* The three phases of histogram (ESD) of eigenvalues of $\mathbf{W}^T\mathbf{W}$ during network training.

small bulk eigenvalues and recovering spiked eigenvalues of $\boldsymbol{\Sigma}_{PUB}$ from $\mathbf{W}^T\mathbf{W}$, thereby improving the generalization performance. A following question is how to determine the boundary between informative spikes and noisy bulk eigenvalues. (Staats et al., 2023) determine the boundary by cross-validation, which is purely data-driven without theoretical support and is quite computationally expensive. (Ke et al., 2023) propose the Bulk Eigenvalue Matching Analysis (BEMA) algorithm based on the PUB model. Using BEMA, a weight matrix pruning algorithm is further developed by (Shmalo et al., 2023). However, the existing compression algorithm either rely on the PUB model or empirical adjustments without any theoretical support.

Regrettably, the assumption of the PUB model is too restrictive, as it requires $\mathbf{W}^T\mathbf{W}$ to have a homogeneous population variance $\sigma^2$. It also fails to accurately capture the eigenvalues distribution of $\mathbf{W}^T\mathbf{W}$ in many empirical studies. Therefore, we consider a more general PDB model to accommodate heterogeneous population variances. Specifically, we propose that during the training process, the initial $\mathbb{E}\mathbf{W}_0^T\mathbf{W}_0 = \sigma_0^2\mathbf{I}_p$ will evolve into $\mathbb{E}\mathbf{W}^T\mathbf{W}$ $= \mathrm{diag}(\underbrace{\alpha_1,\ldots,\alpha_K}_{K\ \textbf{spikes}}, \underbrace{\sigma_1^2,...,\sigma_1^2}_{\textbf{bulk1}}, \underbrace{\sigma_2^2,...,\sigma_2^2}_{\textbf{bulk2}})$. Note that when $\sigma_1^2 = \sigma_2^2$, our PDB model will degenerate to the PUB model. We didn't adopt more general $M$-Bulks ($M \geq 3$) models because we observe that the proportion of additional bulks is negligible and the PDB model has shown sufficiently superior performance than PUB model in Fig. 2.

Based on PDB model, we posit that both the spikes and the bulk1 contain valuable information, whereas the bulk2 predominantly represents noise. We further propose a **Population Double Bulk Least Squares (PDBLS) algorithm** to estimate the structure of $\mathbb{E}\mathbf{W}^T\mathbf{W}$, from which we can determine the boundary between noisy eigenvalues and information. Moreover, as shown in Fig. 5, we develop a new **PDB Noise-Filtering algorithm** to compress the weight matrix by only removing the smaller bulk2 eigenvalues while

keeping the bulk1. The spiked information of $\mathbb{E}\mathbf{W}^T\mathbf{W}$ is also recovered. Experiments demonstrate that our proposed PDB model outperforms the PUB model in fitting the eigenvalues of $\mathbf{W}^T\mathbf{W}$, and our compressed weight matrices exhibit lower rank while preserving test accuracy or even improving it when label contains noise. In summary, our contributions include:

1. We propose a generalized PDB model to characterize the eigenvalue behavior of $\mathbf{W}^T\mathbf{W}$ in the bulk+spikes phase. This model accurately captures the empirical distribution of eigenvalues, thus confirming its validity.

2. We propose an efficient hyperparameter-free algorithm PDBLS to estimate $\mathbb{E}\mathbf{W}^T\mathbf{W}$, which establishes the relationship between the eigenvalues of $\mathbf{W}^T\mathbf{W}$ and $\mathbb{E}\mathbf{W}^T\mathbf{W}$. This enables us to establish a boundary between noise and information and determine how much information of $\mathbf{W}^T\mathbf{W}$ needs to be retained.

3. We introduce a novel PDB Noise-Filtering algorithm to compress the weight matrix by removing noisy eigenvalues and retrieving information from $\mathbb{E}\mathbf{W}^T\mathbf{W}$. Our algorithm also recommends the best compression ratio of weight matrix where further compression will lead to significantly loss of generalization ability. Experiments show that our approach can significantly reduce the rank of weight matrix $\mathbf{W}$ while preserving network generalization performance, and even enhancing it in the presence of noise.

## 2. Motivation: from PUB to PDB model

In this section, we introduce the motivation that why we choose the PDB model.

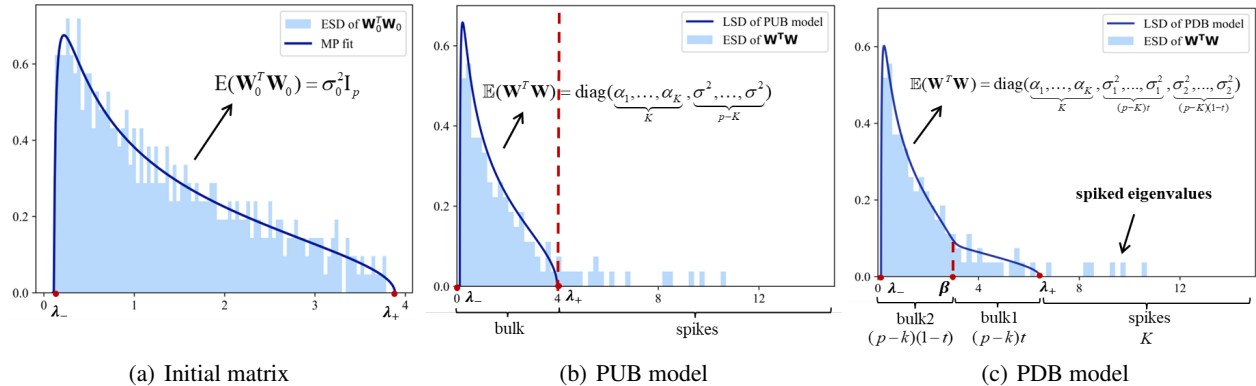

(a) Initial matrix      (b) PUB model      (c) PDB model

*Figure 2.* The histogram (ESD) and LSD of eigenvalues of $\mathbf{W}^T\mathbf{W}$, where (a) shows a initial matrix, (b) and (c) show trained matrix with PUB and PDB model. The vertical dashed lines represent the noise-information boundaries under the two models.

## 2.1. PUB model

Recently, for weight matrix $\mathbf{W} \in \mathbb{R}^{n \times p}$, (Martin & Mahoney, 2021; Staats et al., 2023) use RMT to study the eigenvalue behavior of $\mathbf{W}^T\mathbf{W}$ before and after training. In the initialization phase, the entries of $\mathbf{W}_0$ are i.i.d. generated satisfying $\mathbf{\Sigma}_0 = \mathbb{E}\mathbf{W}_0^T\mathbf{W}_0 = \sigma_0^2\mathbf{I}_p$ and the LSD of $\mathbf{W}_0^T\mathbf{W}_0$ has density function (**MP law**):

$$f(x; c, \sigma_0^2) = \frac{\sqrt{(\lambda_+ - x)(x - \lambda_-)}}{2\pi\sigma_0^2 cx}\mathbb{1}_{\{\lambda_- \leq x \leq \lambda_+\}}.$$

Here the boundary points $\lambda_\pm(\sigma_0^2) = \sigma_0^2(1 \pm \sqrt{c})^2$, $c = \frac{p}{n}$.

(Martin & Mahoney, 2021) point out that after training, the eigenvalues of $\mathbf{W}^T\mathbf{W}$ can be divided into two categories: the bulk eigenvalues and the spiked eigenvalues. As shown in Fig. 2, the bulk eigenvalues are tightly clustered with their histogram conforming to the LSD. On the contrary, the spiked eigenvalues are far from bulk eigenvalues and lie outside the boundaries of LSD. In response to this phenomenon, (Shmalo et al., 2023; Staats et al., 2023) explore a PUB model to characterize the eigenvalues of $\mathbf{W}^T\mathbf{W}$ in the bulk+spikes phase. They assume, after training,

$$\mathbf{\Sigma}_{PUB} = \mathbb{E}\mathbf{W}^T\mathbf{W} = \text{diag}(\underbrace{\alpha_1, \ldots, \alpha_K}_{K}, \underbrace{\sigma^2, \ldots, \sigma^2}_{p-K}).$$

$$(1)$$

Here $\{\alpha_1, \ldots, \alpha_K\}$ are $K$ population spiked eigenvalues and $\{\sigma^2, \ldots, \sigma^2\}$ are $p - K$ bulk eigenvalues of $\mathbf{\Sigma}_{PUB}$. During training process, the initial variance $\sigma_0^2$ changes to $\sigma^2$ and extra $K$ spikes appear. Correspondingly, the LSD of sample bulk eigenvalues of $\mathbf{W}^T\mathbf{W}$ follows the MP law with parameter $\sigma^2$. While the $K$ sample spiked eigenvalues lie outside the boundary of the LSD, i.e., $\sigma^2(1 + \sqrt{c})^2$. The relationship between the sample eigenvalues of $\mathbf{W}^T\mathbf{W}$ and population eigenvalues of $\mathbf{\Sigma}_{PUB}$ is given below.

$$\text{Eig}(\mathbf{\Sigma}_{PUB}) = \{\underbrace{\alpha_1, \ldots, \alpha_K}_{K}, \underbrace{\sigma^2, \ldots, \sigma^2}_{p-K}\}$$
$$\updownarrow \text{ spikes} \qquad \updownarrow \text{ bulk}$$
$$\text{Eig}(\mathbf{W}^T\mathbf{W}) = \{\underbrace{\lambda_1, \ldots, \lambda_K}_{K}, \underbrace{\lambda_+ \geq \lambda_j \geq \lambda_-}_{p-K}\}.$$

## 2.2. PDB model

However, the assumption of the PUB model is quite restrictive, as it requires $\mathbf{W}^T\mathbf{W}$ to have a homogeneous population variance $\sigma^2$. Additionally, the eigenvalue distribution of $\mathbf{W}^T\mathbf{W}$ does not perfectly align with the MP law with $\sigma^2$ under PUB model, as shown in Fig. 2(b). To address these limitations, we consider a more general model which can accommodate heterogeneous population variances. Specifically, we extend $\sigma^2$ to $M$ different positive constants $\{\sigma_i^2, 1 \leq i \leq M\}$ with proportions $\{t_i, 1 \leq i \leq M\}$:

$$\mathbf{\Sigma}_{PUB} = \mathbb{E}\mathbf{W}^T\mathbf{W} = \text{diag}(\alpha_1, \ldots, \alpha_K, \boxed{\sigma^2, \ldots, \sigma^2})$$
$$\Downarrow$$
$$\mathbb{E}\mathbf{W}^T\mathbf{W} = \text{diag}(\underbrace{\alpha_1, \ldots, \alpha_K}_{K}, \boxed{\underbrace{\sigma_1^2, \ldots, \sigma_1^2}_{(p-K)t_1}, \ldots, \underbrace{\sigma_M^2, \ldots, \sigma_M^2}_{(p-K)t_M}}).$$

In empirical studies, we observe that $M = 2$ is sufficient, because the proportion of additional $\sigma_i$ terms is negligible. For example, Table 1 presents the estimated proportions for $M = 4$, where $\hat{t}_3, \hat{t}_4$ are very small. Therefore, in this study,

*Table 1.* The estimated proportion of each bulk when $M = 4$.

| | $\sigma_1^2$ | $\sigma_2^2$ | $\sigma_3^2$ | $\sigma_4^2$ |
|---|---|---|---|---|
| $\hat{t}_i$ | 0.6803 | 0.2719 | 0.0474 | 0.0004 |

we propose a generalized **Population Double Bluk (PDB) model** for trained weight matrix $\mathbf{W}$, satisfying

$$\Sigma_{PDB} := \mathbb{E}\mathbf{W}^T\mathbf{W}$$
$$= \text{diag}(\underbrace{\alpha_1,\ldots,\alpha_K}_{K}, \underbrace{\sigma_1^2,\ldots,\sigma_1^2}_{(p-K)t}, \underbrace{\sigma_2^2,\ldots,\sigma_2^2}_{(p-K)(1-t)}). \quad (2)$$

where $\{\alpha_1,\ldots,\alpha_K\}$ are $K$ population spikes of $\Sigma_{PDB}$ and $t$ is the proportion of $\sigma_1^2$ among all bulk eigenvalues. Similar as PUB model, PDB model also has one-to-one corresponding relationship between the sample eigenvalues of $\mathbf{W}^T\mathbf{W}$ and population eigenvalues of $\Sigma_{PDB}$:

$$\text{Eig}(\Sigma_{PDB}) = \{\underbrace{\alpha_1,\ldots,\alpha_K}_{K}, \quad \underbrace{\sigma_1^2,\ldots,\sigma_1^2}_{(p-K)t}, \quad \underbrace{\sigma_2^2,\ldots,\sigma_2^2}_{(p-K)(1-t)}\}$$
$$\Updownarrow \text{ spikes} \qquad \Updownarrow \text{ bulk1} \qquad \Updownarrow \text{ bulk2}$$
$$\text{Eig}(\mathbf{W}^T\mathbf{W}) = \{\underbrace{\lambda_1,\ldots,\lambda_K}_{K}, \quad \underbrace{\lambda_+ \geq \lambda_j \geq \beta}_{(p-K)t}, \quad \underbrace{\beta > \lambda_j \geq \lambda_p}_{(p-K)(1-t)}\}$$

where $\beta = \lambda_{K+(p-K)t}$ is the boundary point between sample bulk1 and bulk2 of $\mathbf{W}^T\mathbf{W}$. **Note that the PUB model** (1) **is a special case of our PDB model when** $t = 0$.

**More importantly, the proposed PDB model demonstrates significantly better empirical performance than the PUB model.** Firstly, the density curve of LSD under the PDB model aligns more close to the histogram of $\mathbf{W}^T\mathbf{W}$, see Fig. 3. The PDB model parameters for Fig. 3 are listed in Table 2.

*Table 2.* The results of $\Theta_{bulk} = \{\sigma_1^2, \sigma_2^2, t\}$ for the DNN.

| Estimator | $\hat{t}$ | $\hat{\sigma}_1^2$ | $\hat{\sigma}_2^2$ |
|---|---|---|---|
| FCNN: MNIST | 0.17 | 3.37 | 1.36 |
| VGG16: CIFAR10 | 0.25 | 1.61 | 0.75 |
| ResNet18:ImageNet | 0.35 | 7.54 | 0.96 |
| ResNet18: CIFAR10 | 0.25 | 5.06 | 0.95 |

Secondly, our PDB model achieves superior alignment with the spectral moments of $\mathbf{W}^T\mathbf{W}$ compared to the PUB model. Specifically, we employ different methods to estimate model parameters and compare the first three theoretical moments with the empirical values, including the BEMA method (Ke et al., 2023) and the Kernel approach (Staats et al., 2023) for the PUB model, and the PDBLS algorithm for the PDB model (see Section 3.2). Detailed results are presented in Tables 3–5, where Table 3-4 correspond to the FCNN and convolution networks, and Table 5 presents the large language models. The theoretical spectral moment formulas can be found in the Appendix. It's clear that our PDB model matches the empirical moments much better than PUB model, especially higher order moments.

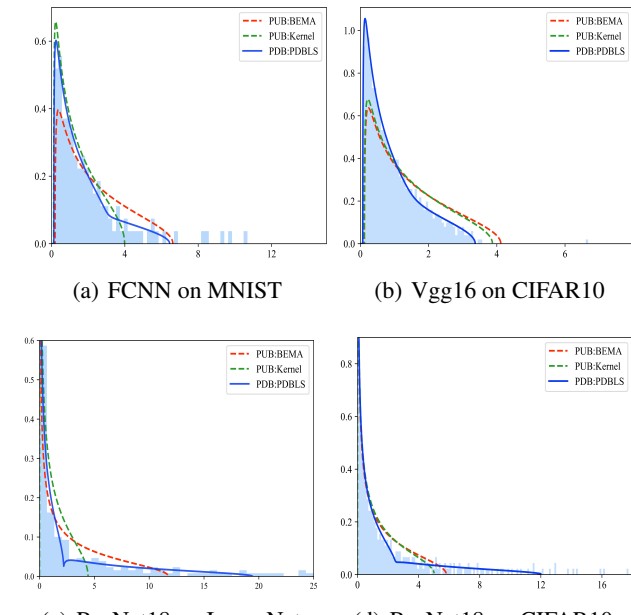

(a) FCNN on MNIST      (b) Vgg16 on CIFAR10

(c) ResNet18 on ImageNet     (d) ResNet18 on CIFAR10

*Figure 3.* Histograms of the eigenvalues of $\mathbf{W}^T\mathbf{W}$ and density curves of LSDs based on different models. The solid blue line represents PDB model, and the dashed red and green curves are for PUB model.

*Table 3.* Comparison of theoretical and empirical spectral moments for FCNN and VGG16, $\hat{\gamma}_j = \frac{1}{p}\text{tr}(\mathbf{W}^T\mathbf{W})^j$, $j = 1, 2, 3$.

| | | FCNN:MNIST | | | VGG16:CIFAR10 | | |
|---|---|---|---|---|---|---|---|
| Model | Method | $\gamma_1$ | $\gamma_2$ | $\gamma_3$ | $\gamma_1$ | $\gamma_2$ | $\gamma_3$ |
| PUB | BEMA | 2.27 | 7.74 | 32.23 | 0.92 | 1.26 | 2.12 |
| PUB | Kernel | 1.37 | 2.81 | 7.04 | 0.90 | 1.22 | 2.03 |
| **PDB** | **PDBLS** | **1.71** | **4.94** | **18.79** | **0.96** | **1.53** | **3.11** |
| empirical $\hat{\gamma}_j$ | | 1.74 | 5.50 | 24.17 | 0.97 | 1.59 | 3.51 |

*Table 4.* Comparison of theoretical and empirical spectral moments for FCNN and ResNet18.

| | | ResNet18:ImageNet | | | ResNet18:CIFAR10 | | |
|---|---|---|---|---|---|---|---|
| Model | Method | $\gamma_1$ | $\gamma_2$ | $\gamma_3$ | $\gamma_1$ | $\gamma_2$ | $\gamma_3$ |
| PUB | BEMA | 2.94 | 17.26 | 126.79 | 1.80 | 4.82 | 15.84 |
| PUB | Kernel | 1.11 | 2.45 | 6.77 | 1.83 | 5.02 | 16.84 |
| **PDB** | **PDBLS** | **3.23** | **30.62** | **377.50** | **2.27** | **9.14** | **49.08** |
| empirical $\hat{\gamma}_j$ | | 3.32 | 31.48 | 402.79 | 2.28 | 9.38 | 52.65 |

## 3. PDB estimation

In this section, we provide a detailed characterization of the asymptotic properties of the sample eigenvalues of $\mathbf{W}^T\mathbf{W}$, including the bulk eigenvalues, bulk boundaries, and spiked eigenvalues. Building on these properties, we propose the **PDBLS algorithm** to estimate the model parameters of

*Table 5.* Comparison of theoretical and empirical spectral moments for T5-base and BERT.

| Model | Method | T5-base: RTE | | | BERT: SCITAIL | | |
|---|---|---|---|---|---|---|---|
| | | $\gamma_1$ | $\gamma_2$ | $\gamma_3$ | $\gamma_1$ | $\gamma_2$ | $\gamma_3$ |
| PUB | BEMA | 0.67 | 0.90 | 1.51 | 0.59 | 0.69 | 1.02 |
| PUB | Kernel | 0.53 | 0.56 | 0.74 | 0.56 | 0.63 | 0.88 |
| **PDB** | **PDBLS** | **0.77** | **1.55** | **4.17** | **0.67** | **1.18** | **2.78** |
| empirical $\hat{\gamma}_j$ | | 0.72 | 1.83 | 5.35 | 0.71 | 1.38 | 3.95 |

$\mathbf{\Sigma}_{PDB}$, which allows us to determine the boundary between noise and information.

### 3.1. Sample eigenvalue behavior of PDB model

The following three theorems illustrate the relationship between sample eigenvalues of $\mathbf{W}^T\mathbf{W}$ and population parameters $\{K, \alpha_1, \ldots, \alpha_K, \sigma_1^2, \sigma_2^2, t\}$ of $\mathbf{\Sigma}_{PDB}$ .

**Theorem 3.1.** *(sample bulk eigenvalues) Under PDB model (2), as $n \to \infty$, $p/n \to c$, the empirical spectral distribution of $\mathbf{W}^T\mathbf{W}$, $F_n^{\mathbf{W}^T\mathbf{W}}(x) = \frac{1}{p}\sum_{j=1}^p \mathbb{1}_{\{\lambda_j \leq x\}}$ a.s. converges to the LSD with the density function:*

$$\rho(x) = \lim_{\eta \to 0} \frac{\operatorname{Im}\underline{m}(z)}{\pi c}, \quad z = x + i\eta, \quad \eta > 0, \quad (3)$$

*where $\operatorname{Im}\underline{m}(z)$ represents the imaginary part of $\underline{m}(z)$ and $\underline{m}(z)$ satisfies the following equation:*

$$z = -\frac{1}{\underline{m}(z)} + \frac{ct\sigma_1^2}{1 + \sigma_1^2\underline{m}(z)} + \frac{c(1-t)\sigma_2^2}{1 + \sigma_2^2\underline{m}(z)}. \quad (4)$$

An example of $\rho(x)$ is shown in Fig. 2(c) where $\{\sigma_1^2 = 4.63, \sigma_2^2 = 1.67, t = 0.20\}$.

**Theorem 3.2.** *(sample boundaries) The boundary between sample spikes and bulk1 of $\mathbf{W}^T\mathbf{W}$ is*

$$\lambda_+ = g(y), \ y = \arg\max_{x \in \mathbb{R}}\{g'(x) = 0\}, \quad (5)$$

$$g(x) = x + cx\frac{t\sigma_1^2}{x - \sigma_1^2} + cx\frac{(1-t)\sigma_2^2}{x - \sigma_2^2}. \quad (6)$$

*The boundary between sample bulk1 and bulk2 of $\mathbf{W}^T\mathbf{W}$ is*

$$\beta = \lambda_{K+pt-Kt}.$$

**Theorem 3.3.** *(sample spiked eigenvalues) Under PDB model (2), as $n \to \infty$, $p/n \to c$, the sample spiked eigenvalues $\{\lambda_1, \ldots, \lambda_K\}$ of $\mathbf{W}^T\mathbf{W}$ converges to functions of population spiked eigenvalues $\{\alpha_1, \ldots, \alpha_K\}$ of $\mathbf{\Sigma}_{PDB}$:*

$$\lambda_j \xrightarrow{a.s.} g(\alpha_j), \quad j \in \{1, \ldots, K\}. \quad (7)$$

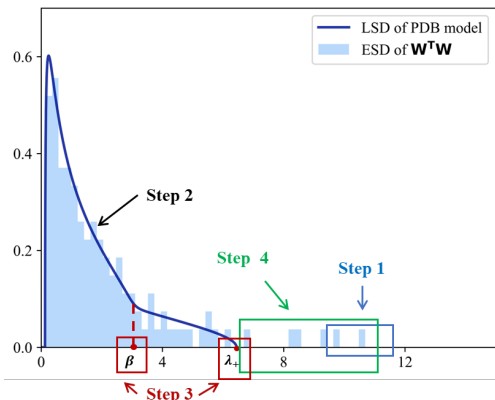

*Figure 4.* The histogram (ESD) of eigenvalues and LSD of PDB model. The estimation procedure (PDBLS Algorithm) for $\Theta_{bulk} = \{t, \sigma_1^2, \sigma_2^2\}$, $\Theta_{spike} = \{K, \alpha_1, \ldots, \alpha_K\}$ and $\Theta_{bound} = \{\lambda_+, \beta\}$ are also illustrated in this figure.

### 3.2. PDBLS algorithm: Estimation of PDB model

Based on Theorems 3.1-3.3, we introduce an algorithm called **Population Double Bulk Least Squares (PDBLS)** to estimate $\mathbf{\Sigma}_{PDB} = \mathbb{E}\mathbf{W}^T\mathbf{W}$ and two sample boundary points: one is the boundary between the sample spikes and the bulk1, $\lambda_+$, and the other is the boundary between bulk1 and bulk2, $\beta$. Our goal is to estimate the bulk parameters $\Theta_{bulk} = \{\sigma_1^2, \sigma_2^2, t\}$, the spike parameters $\Theta_{spike} = \{K, \alpha_1, \ldots, \alpha_K\}$, and the boundary parameters $\Theta_{bound} = \{\lambda_+, \beta\}$. We denote these estimators as $\{\hat{\sigma}_1^2, \hat{\sigma}_2^2, \hat{t}\}$, $\{\hat{K}, \hat{\alpha}_1, \ldots, \hat{\alpha}_{\hat{K}}\}$ and $\{\hat{\lambda}_+, \hat{\beta}\}$. We start with the estimation of $\Theta_{bulk}$ and $\Theta_{bound}$. Although spiked eigenvalues don't affect LSD, some exceptionally large spikes can result in an overestimation of bulk parameters in the finite sample case. As depicted in the Fig. 4, there are several extremely large eigenvalues highlighted in the blue box, indicating the necessity for us to give an initial estimation of the number of spikes $K_0$. By mitigating the influence of the top $K_0$ spikes, we provide estimates for $\Theta_{bulk}$ first, followed by $\Theta_{bound}$, and further refine the number and values of spikes in $\Theta_{spike}$. In detail, there are four main steps to obtain these estimators.

**Step 1 Initially estimate the number of spikes $K_0$.**

Inspired by (Liu et al., 2023), the location that corresponds to the smallest distance between the consecutive eigenvalues can be used to determine the number of spikes. The initial number of spikes based on this criteria in (Liu et al., 2023) can be calculated by:

$$\hat{K}_0 = \arg\min \frac{1}{n}[-n(\lambda_1 - \lambda_{k+1}) + n(p-k-1)\log\tilde{\theta}_{p,k} + 2pk],$$
$$(8)$$

where $\tilde{\theta}_{p,k} = \frac{1}{p-k-1}\sum_{i=k+1}^{p-1}\exp\{2(\lambda_i - \lambda_{i+1})\}$ and $\omega = \lfloor 6n^{0.1}\rfloor$, $\lfloor\cdot\rfloor$ represents the floor function. Then we remove

---

**Algorithm 1 PDBLS algorithm (For estimation)**

---

**Input**: Eigenvalues $\lambda_j$ of matrix $\mathbf{W}^T\mathbf{W}$.

**Output**: $\widehat{\Theta}_{bulk} = \{\hat{\sigma}_1^2, \hat{\sigma}_2^2, \hat{t}\}$, $\widehat{\Theta}_{spike} = \{\hat{K}, \hat{\alpha}_1, \ldots, \hat{\alpha}_{\hat{K}}\}$, $\widehat{\Theta}_{bound} = \{\hat{\lambda}_+, \hat{\beta}\}$.

    *// **Initially estimate** $K_0$*

1: Roughly estimate the number of spikes $\hat{K}_0$ by Eq. (8);
2: Remove the top $\hat{K}_0$ spikes;

    *// **Estimate** $\Theta_{bulk}$*

3: Calculate $\hat{u}_i(\Theta_{bulk})$ by Eq. (10);
4: Minimize Eq. (9) to solve $(\sigma_1^2, \sigma_2^2, t)$;

    *// **Estimate** $\Theta_{bound}$ **and Refine** $K$*

5: Calculate boundary point $\hat{\lambda}_+$ by Eq. (5);
6: Estimate the number of spikes $\hat{K}$ by Eq. (11);
7: Estimate boundary point $\hat{\beta}$ via Eq. (12).

    *// **Estimate** $\Theta_{spike}$*

8: Recover population spikes $\{\alpha_j\}_{j=1}^{\hat{K}}$ from sample spikes by Eq. (6) and Eq. (7);

---

the top $\hat{K}_0$ spikes which may affect the estimation of $\Theta_{bulk}$.

**Step 2 Estimate** $\Theta_{bulk} = \{\sigma_1^2, \sigma_2^2, t\}$.

The bulk parameters can be estimated by solving the following least squares optimization problem (Li et al., 2013):

$$\arg\min_{\Theta_{bulk}} \sum_{i=1}^{q} (\hat{u}_i(\Theta_{bulk}) - u_i)^2, \tag{9}$$

where $\{u_1, ..., u_q\}$ take equally spaced $j$-points in each interval of $\mathcal{U}$. Here

$$\mathcal{U} = \begin{cases} (-10, 0) \cup (0, 0.5\lambda_{min}) \cup (5\lambda_{max}, 10\lambda_{max}), p \neq n. \\ (-10, 0) \cup (5\lambda_{max}, 10\lambda_{max}), p = n. \end{cases}$$

We set $j = 20$ in all experiments of Section 5.

According to (Yao et al., 2015) and Eq. (4) in Section 2, for each $u_i$, we can calculate $\hat{u}_i(\Theta_{bulk})$ as:

$$\hat{u}_i(\Theta_{bulk}) = -\frac{1}{\underline{m}_n(u_i)} + \frac{p - \hat{K}_0}{n} \frac{t\sigma_1^2}{1 + \sigma_1^2 \underline{m}_n(u_i)} \\ + \frac{p - \hat{K}_0}{n} \frac{(1 - t)\sigma_2^2}{1 + \sigma_2^2 \underline{m}_n(u_i)}, \tag{10}$$

where $\underline{m}_n(u_i) = -\frac{1 - (p - \hat{K}_0)/n}{u_i} + \frac{1}{n} \sum_{l=1}^{p - \hat{K}_0} \frac{1}{\lambda_l - u_i}$.

By solving (9), we obtain the bulk estimators $\widehat{\Theta}_{bulk} = (\hat{\sigma}_1^2, \hat{\sigma}_2^2, \hat{t})$. Subsequently, the density curve of LSD of $\mathbf{W}^T\mathbf{W}$ naturally follows from Eq. (3)-(4).

**Step 3 Estimate** $\Theta_{bound} = \{\lambda_+, \beta\}$ **and refine the number of spikes** $K$**.**

From Fig. 2(c), we can see that the number of spikes is equal to

$$K = \#\{\lambda_j | \lambda_j \in (\lambda_+, \lambda_{max}]\}, \tag{11}$$

where $\#\{\cdot\}$ represents the cardinality of the set. With the estimated $\widehat{\Theta}_{bulk}$ in Step 2, we can obtain the estimator of boundary point $\hat{\lambda}_+$ via Eq.(5). Then we can further refine the estimation of the number of spikes $\hat{K}$ via Eq. (11). Subsequently according to Theorem 3.2, we estimate $\beta$ by

$$\hat{\beta} = \lambda_{\hat{K} + p\hat{t} - \hat{K}\hat{t}}. \tag{12}$$

**Step 4 Estimate** $\Theta_{spike} = \{\alpha_1, ...\alpha_{\hat{K}}\}$.

The relationship between spikes of $\mathbf{W}^T\mathbf{W}$ and $\mathbf{\Sigma}_{PDB}$ is established in Theorem 3.3. Thus we can obtain $\{\hat{\alpha}_1, \ldots, \hat{\alpha}_{\hat{K}}\}$ by Eq. (6) and (7).

In summary, **the estimation algorithm is thoroughly detailed in Algorithm 1**. Based on Theorems 3.1-3.3, we can show that PDBLS is a consistent estimator of $\mathbf{\Sigma}_{PDB}$.

**Theorem 3.4.** *Under PDB model* (2), *as* $n \to \infty, p/n \to c$,

$$\{\widehat{\Theta}_{bulk}, \widehat{\Theta}_{spike}, \widehat{\Theta}_{bound}\} \xrightarrow{a.s.} \{\Theta_{bulk}, \Theta_{spike}, \Theta_{bound}\}.$$

# 4. Matrix compression of PDB model

In this section, we determine the boundary between noise and information and introduce a novel PDB Noise-Filtering algorithm to compress weight matrices .

Now with the estimation of our PDB model, we have a clear picture about two boundaries of $\mathbf{W}$, one is the boundary between the sample spikes and the bulk1, $\lambda_+$, and the other is the boundary between bulk1 and bulk2, $\beta$. This sheds new light on determining the noise-information boundary, which facilitates the compression of the weight matrix.

In the current literature, existing method to determine the boundary between noise and information are either purely data-driven or based on the PUB model. As for the PUB model, (Shmalo et al., 2023) and (Staats et al., 2023) treat the MP edge $\lambda_+$ of LSD, the vertical dashed line in Fig. 2(b), as the noise-information boundary. They assume that all the information is contained in the spikes of $\mathbf{W}^T\mathbf{W}$ while bulk eigenvalues only represent noise. Accordingly, in their compression algorithm, they remove all the bulk singular values and recover the population spiked singular values from sample spikes of weight matrices. However, the boundary inferred from the PUB model is quite stringent. Consequently, the corresponding compression algorithm would lead to a substantial loss of useful information. As a result, the compressed network would struggle to maintain the original generalization performance.

Therefore, to strike a balance between the amount of information retained and the degree of weight matrix rank

**Algorithm 2 PDB Noise-Filtering algorithm (For matrix compression)**

---

1: Perform the singular value composition of the weight matrix $\mathbf{W} = \mathbf{U}\mathbf{D}^{\frac{1}{2}}\mathbf{V}$, where $\mathbf{D} = \mathrm{diag}(\lambda_1, \ldots, \lambda_p)$.

2: Apply the PDBLS algorithm to estimate PDB model including $\widehat{\Theta}_{bulk} = \{\hat{\sigma}_1^2, \hat{\sigma}_2^2, \hat{t}\}$, $\widehat{\Theta}_{spike} = \{\hat{K}, \hat{\alpha}_1, \ldots \hat{\alpha}_{\hat{K}}\}$ and $\widehat{\Theta}_{bound} = \{\hat{\lambda}_+, \hat{\beta}\}$.

3: Replace sample spikes $\{\lambda_j \in (\hat{\lambda}_+, \lambda_{max}]\}$ with population spikes $\{\hat{\alpha}_1, \ldots, \hat{\alpha}_{\hat{K}}\}$, retain the bulk1 $\{\lambda_j \in [\hat{\beta}, \hat{\lambda}_+]\}$ in $\mathbf{D}$, and replace the bulk2 $\{\lambda_j \in [\lambda_{min}, \hat{\beta})\}$ with zero. The new diagonal matrix is formulated as

$$\mathbf{D}_{new} = \mathrm{diag}\left\{\{\hat{\alpha}_1, \ldots, \hat{\alpha}_{\hat{K}}\}, \{\lambda_j \in [\hat{\beta}, \hat{\lambda}_+]\}, \mathbf{0}\right\}.$$

4: Obtain a new weight matrix $\mathbf{W}_{new} = \mathbf{U}\mathbf{D}_{new}^{\frac{1}{2}}\mathbf{V}$, whose rank is $\hat{K} + (p - \hat{K})\hat{t}$.

---

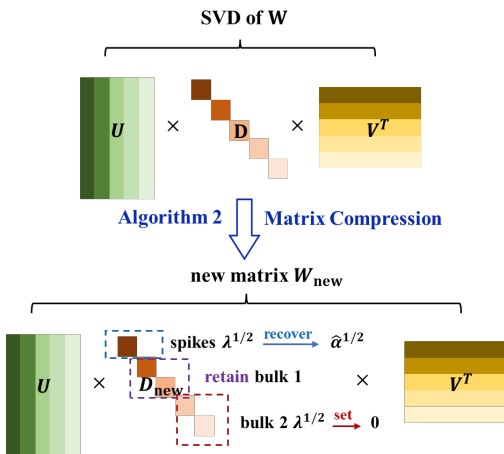

*Figure 5.* The process of matrix compression

reduction, we extend the PUB model to the PDB model, where only one bulk represents noise, while all the trained information is retained in all the spikes and the other bulk. In other words, **our proposed noise-information boundary is the boundary between sample bulk1 and bulk2**, $\beta$, the vertical dashed line in Fig. 2(c). In this way, we retain more important eigenvalues of $\mathbf{W}^T\mathbf{W}$ and find a more suitable way to recover the important information in $\Sigma_{PDB}$. The detailed process of matrix compression is given in Fig. 5. Specifically, with the estimated $\Sigma_{PDB}$, we remove all the smaller eigenvalues in the bulk2 of $\mathbf{W}^T\mathbf{W}$ while keeping the bulk1 as a significant source of information. Simultaneously, we incorporate the recovered population spikes $\widehat{\Theta}_{spike}$ as another valuable source of information learned from $\mathbf{W}$ during the training process. **The detailed compression algorithm is outlined in Algorithm 2.**

# 5. Experiments

In this section, we conduct numerical experiments to demonstrate the superiority of our PDB model and the effectiveness of the weight matrix compression algorithm.

## 5.1. Experimental setups

We evaluate generalization performance using test accuracy and employ three basic neural network architectures, the three-layer Fully Connected Neural Network (FCNN), Residual Network-18 (ResNet18) (He et al., 2015), and Visual Geometry Group-16 (VGG16) (Simonyan & Zisserman, 2015) . The FCNN is trained on MNIST, while ResNet18 and VGG16 are evaluated on CIFAR10 (Krizhevsky et al., 2009) and ImageNet (Deng et al., 2009). The three architectures configurations including the following:

1. **FCNN trained on MNIST.**

   The FCNN for MNIST is fc784 → fc512 → fc512 → fc350 → fc10, where fc denotes fully connected layer.

2. **ResNet18 trained on CIFAR10 and ImageNet.**

   We add a linear layer of size $512 \times 256$ before the linear layer in the original ResNet18 when training CIFAR10.

3. **VGG16 trained on CIFAR10 and ImageNet.**

   The size of last two linear layers is changed from 4096 to 256 when training CIFAR10.

Additionally, we assess the generalization of three representative pre-trained architectures: BERT (Devlin et al., 2019) and T5-base (Raffel et al., 2020) for natural language processing, and ViT-L (Dosovitskiy, 2020) for computer vision. ViT-L is a variant of the CLIP vision encoder (Radford et al., 2021). In the CLIP model, the text encoder remains frozen, and text embeddings are generated by processing the class labels through it. For the language models, experiments are conducted on the RTE (Wang et al., 2018) and SciTail (Khot et al., 2018) datasets, while the vision model is tested on DTD (Cimpoi et al., 2014) and SUN397 (Xiao et al., 2016).

Each image is normalized to the range of [0,1], and the weight matrices of the networks are initialized by the Glorot uniform distribution (Glorot & Bengio, 2010). For basic architectures, we employ SGD with an exponential decay learning rate during the training phase. The activation functions used are $\mathrm{Relu}(\cdot)$ for hidden layers and $\mathrm{Softmax}(\cdot)$ for output layer. For large-scale model, we utilize the AdamW (Loshchilov & Hutter, 2017) optimizer in conjunction with a cosine learning rate scheduler. Regarding convolutional layers, we follow the scheme in (Idelbayev & Carreira-Perpinán, 2020) for reshaping the convolution kernel into a 2D matrix. In particular, since a convolution layer has $n$

groups of convolutional kernels with $c$ channels, each of size $d \times d$, we can reshape them into a 2D matrix of size $nd \times cd$ for compression. After compression, we can reshape them back to their original 4D convolutional kernel form.

We compare two classes of matrix compression methods: the PUB-based methods and SVD-based methods.

PUB-based methods include:

1. Bulk Eigenvalue Matching Analysis (BEMA) (Ke et al., 2023): find the boundary of density curves by quantile fitting.

2. Kernel Estimation (Kernel) (Staats et al., 2023): use Gaussian broadening to fit MP distribution and find the boundary.

SVD-based methods include:

1. Sparse low rank (SLR) (Swaminathan et al., 2020) : obtain low-rank matrices by imposing rank-sparsity constraints.

2. Naive SVD (Shmalo et al., 2023): set 45% small singular values to zero.

### 5.2. Generalization and compression performance

In this section, we train the network on data sets with and without noisy labels, shuffling 60% of the labels to introduce noise. **Then, we compress networks according to Algorithm 2 and Fig. 5.** The test accuracy before and after compression for different methods is compared to evaluate the generalization performance of compressed networks. We denote the test accuracy before compression as the **Base**. All codes in the experiment are conducted on the server equipped with NVIDIA L40 GPUs and Ubuntu 22.04.

Our algorithm will recommend the best compression ratio of weight matrix where further compression will lead to significantly loss of generalization ability. Table 6 lists the rank of the weight matrix before and after compression for various neural networks under four algorithms across different datasets. Fig. 6 reports the test accuracy of various compression algorithms for the dataset with and without label noise. From these results, we obtain the following conclusions:

1. Compared with the original neural network, our compressed model significantly reduces the rank of weight matrices while maintaining its test accuracy. In case of contaminated labels, our compression methods can even improve the test accuracy.

2. Compared with other compression algorithms, our algorithm can achieve better test accuracy at the same

compression level. The boundary point obtained by the PDB model retains more important information.

3. The empirical compression ratio 0.55 proposed in (Shmalo et al., 2023) does not fully compress the weight matrix. On the contrary, the noise-information boundary identified by our method accurately captures the point at which test accuracy starts to decrease during compression.

Table 7 presents the test accuracy of various neural networks when compressed using different algorithms. **Our method achieves the best accuracy, with a maximum improvement of 5%** ($\frac{0.7536}{0.7174} - 1$) **for T5-base: RTE. More comparable outcomes are listed in the Appendix**. These results demonstrate the effectiveness of our proposed method.

Table 8 compares the computational efficiency of four methods, with the time measured in seconds for the total execution and inference phases following data processing. Among them, the computational overhead of our PDB model is slightly higher than that of PUB. However, this additional cost is relatively modest and is outweighed by the improvements in compression performance and model accuracy.

*Table 8.* The computation time of different networks across various comparable algorithms.

| Network | PDB | PUB | Naive SVD | SLR |
|---------|------|------|-----------|------|
| ResNet | 19 | 15 | 18 | 23 |
| VGG16 | 42 | 41 | 47 | 53 |
| BERT | 20 | 15 | 18 | 29 |
| T5-base | 47 | 44 | 45 | 42 |
| VIT-L | 236 | 231 | 243 | 256 |
| Average | **72.8** | 69.2 | 74.2 | 80.6 |

## 6. Conclusions

In this work, we propose a novel Population Double Bulk (PDB) model which can more accurately characterize the singular value behavior of weight matrix than the existing PUB model. An effective PDBLS algorithm is further developed for model estimation and determination of the noise-information boundary. Subsequently, we propose a PDB noise-filtering algorithm to compress weight matrices. Our algorithms will also recommend the best compression ratio of weight matrix where further compression will lead to significantly loss of generalization ability. Experiments demonstrate superiority of our PDB model and effectiveness of our compression method.

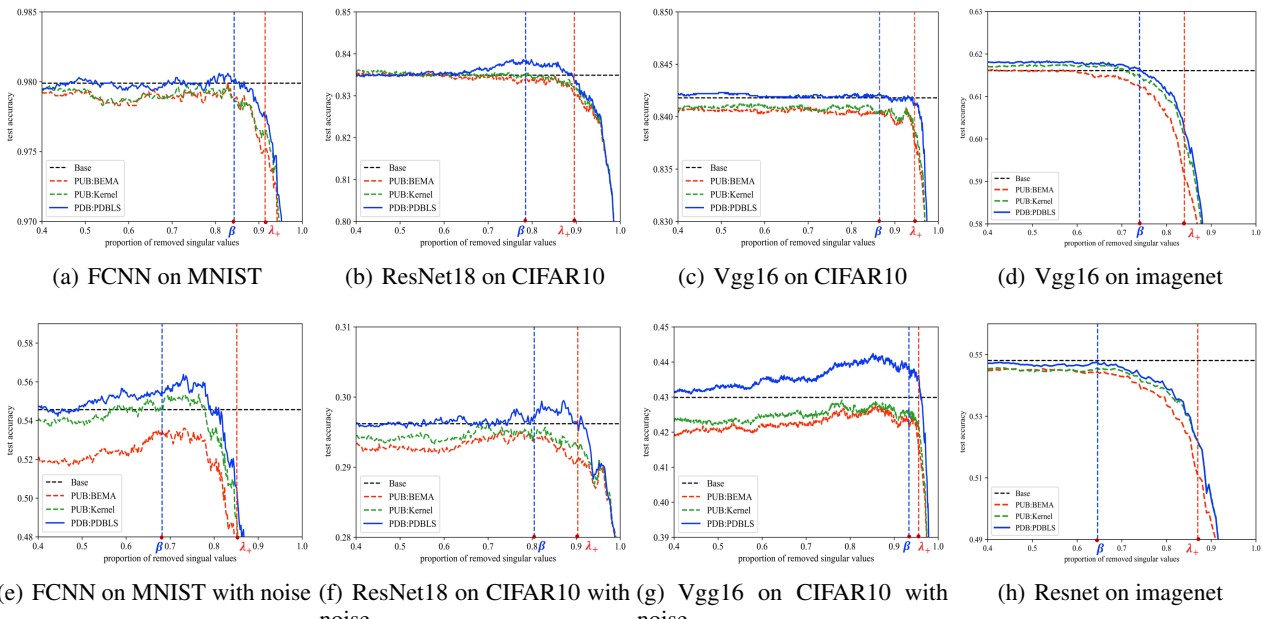

(a) FCNN on MNIST    (b) ResNet18 on CIFAR10    (c) Vgg16 on CIFAR10    (d) Vgg16 on imagenet

(e) FCNN on MNIST with noise   (f) ResNet18 on CIFAR10 with noise   (g) Vgg16 on CIFAR10 with noise   (h) Resnet on imagenet

*Figure 6.* The vertical axis represents test accuracy and the horizontal axis represents the proportion of removed singular values of $\mathbf{W}$ in ascending order. The horizontal dashed black line reports test accuracy before compression. The vertical dashed lines represent the compression ratios recommended by different models, where red $\lambda_+$ for the PUB model and blue $\beta$ for our PDB model.

*Table 6.* Comparison of compression ratios across different compression algorithms, where the ratio equals the quotient of rank of matrix after and before compression.

| Model | FCNN | ResNet18 | | VGG16 | |
|---|---|---|---|---|---|
| Datasets (0% noise) | MNIST | CIFAR10 | ImageNet | CIFAR10 | ImageNet |
| PDB($\beta$) | 55/350 (15.72%) | 165/768 (21.48%) | 68/192 (35.42%) | 208/1536 (13.54%) | 200/768 (26.04%) |
| PUB($\lambda_+$) | 30/350 (8.57%) | 80/768 (10.42%) | 25/192 (13.02%) | 84/1536 (5.47%) | 123/768 (16.02%) |
| SLR | 80/350 (22.86%) | 200/768 (26.04% ) | 100/192 (52.08%) | 200/1536 (13.02%) | 260/768 (33.85%) |
| Naive SVD | 192/350 (55%) | 422/768 (55%) | 106/192 (55%) | 845/1536 (55%) | 422/768 (55%) |

*Table 7.* The test accuracy obtained by training different network models of different data sets (0% noise) under the four compression methods. **Base** denotes the test accuracy before compression. SLR performs poorly on VIT-L (accuracy 0.1), thus omitted.

| Network | Datasets | **Base** | PDB | PUB | SLR | naive SVD |
|---|---|---|---|---|---|---|
| FCNN | MNIST | 0.9799 | **0.9804** | 0.9791 | 0.9799 | 0.9799 |
| ResNet18 | CIFAR10 | 0.8349 | **0.8384** | 0.8338 | 0.8357 | 0.8354 |
| VGG16 | CIFAR10 | 0.8418 | **0.8422** | 0.8405 | 0.8415 | 0.8419 |
| BERT | RTE | 0.7029 | **0.7319** | 0.7174 | 0.7246 | 0.7029 |
| | SciTail | 0.9055 | **0.9155** | 0.9130 | 0.9008 | 0.9055 |
| T5-base | RTE | 0.7174 | **0.7536** | 0.7319 | 0.7174 | 0.7174 |
| | SciTail | 0.9025 | **0.9243** | 0.9167 | 0.9182 | 0.9196 |
| VIT-L | DTD | 0.7452 | **0.7533** | 0.7482 | - | 0.7405 |
| | SUN397 | 0.7680 | **0.7771** | 0.7720 | - | 0.7716 |
| Average | | 0.7783 | **0.7891** | 0.7827 | 0.7858 | 0.7789 |

# Acknowledgments

This research is supported by National Nature Science Foundation of China NO. 12471258, NO. 12031005 and National Key R&D Program of China(2023YFA1011400). The authors are grateful to the reviewers for their thoughtful comments and suggestions.

## Impact Statement

This paper presents work whose goal is to advance the field of Machine Learning. There are many potential societal consequences of our work, none of which we feel must be specifically highlighted here.

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

# A. Experimental results

Table 9 summarizes the test accuracy achieved by training various network models on various data sets using the four compression techniques. We can further show that our compression algorithm is validated by the outcomes.

*Table 9.* The test accuracy obtained by training different network models of different data sets under the four compression methods. **Base** denotes the test accuracy before compression.

| network | dataset | noise | Base | PDB | PUB | SLR | naive SVD |
|---|---|---|---|---|---|---|---|
| fc784 → fc512 → fc512 → fc350 → fc10 | MNIST | 0% | 0.9799 | **0.9804** | 0.9791 | 0.9799 | 0.9799 |
| | | 10% | 0.9414 | **0.9454** | 0.9418 | 0.9439 | 0.9428 |
| | | 20% | 0.8921 | **0.8968** | 0.8901 | 0.8918 | 0.8906 |
| | | 30% | 0.8013 | **0.8078** | 0.7934 | 0.7893 | 0.8033 |
| | | 40% | 0.7247 | **0.7319** | 0.7119 | 0.7304 | 0.7278 |
| | | 50% | 0.6520 | **0.6551** | 0.6381 | 0.6531 | 0.6387 |
| | | 60% | 0.5455 | **0.5544** | 0.5331 | 0.5536 | 0.5401 |
| Resnet18 | CIFAR10 | 0% | 0.8349 | **0.8384** | 0.8338 | 0.8357 | 0.8354 |
| | | 10% | 0.7360 | **0.7446** | 0.7439 | 0.7391 | 0.7382 |
| | | 20% | 0.6701 | **0.6845** | 0.6831 | 0.6747 | 0.6709 |
| | | 30% | 0.6417 | **0.6576** | 0.6399 | 0.6187 | 0.6466 |
| | | 40% | 0.4995 | **0.5051** | 0.5019 | 0.5027 | 0.5033 |
| | | 50% | 0.4533 | **0.4607** | 0.4597 | 0.4496 | 0.4523 |
| | | 60% | 0.2960 | **0.2982** | 0.2942 | 0.2975 | 0.2960 |
| VGG16 | CIFAR10 | 0% | 0.8418 | **0.8422** | 0.8405 | 0.8415 | 0.8419 |
| | | 10% | 0.7882 | **0.7937** | 0.7931 | 0.7935 | 0.7901 |
| | | 20% | 0.7474 | **0.7496** | 0.7452 | 0.7449 | 0.7478 |
| | | 30% | 0.6992 | **0.7048** | 0.6862 | 0.7011 | 0.6998 |
| | | 40% | 0.6276 | **0.6341** | 0.6255 | 0.6303 | 0.6298 |
| | | 50% | 0.5986 | **0.6122** | 0.6025 | 0.6048 | 0.6024 |
| | | 60% | 0.5570 | **0.5686** | 0.5498 | 0.5591 | 0.5576 |
| Resnet18 | Imagenet | 0% | **0.5473** | 0.5469 | 0.5440 | 0.5467 | 0.5452 |
| VGG16 | Imagenet | 0% | 0.6160 | **0.6164** | 0.6126 | 0.6078 | 0.6076 |
| BERT | RTE | 0% | 0.7029 | **0.7319** | 0.7174 | 0.7246 | 0.7029 |
| | SciTail | 0% | 0.9055 | **0.9155** | 0.9130 | 0.9008 | 0.9055 |
| T5-base | RTE | 0% | 0.7174 | **0.7536** | 0.7319 | 0.7174 | 0.7174 |
| | SciTail | 0% | 0.9025 | **0.9243** | 0.9167 | 0.9182 | 0.9196 |
| VIT-L | DTD | 0% | 0.7452 | **0.7533** | 0.7482 | - | 0.7405 |
| | SUN397 | 0% | 0.7680 | **0.7771** | 0.7720 | - | 0.7716 |

# B. Spectral moment of PDB model

The spectral moment of a matrix is an important statistic related to its eigenvalues, often used to describe the spectral distribution of the matrix. Denote the empirical spectral moment of $\mathbf{W}^T\mathbf{W}$,

$$\hat{\gamma}_j = \frac{1}{p} \operatorname{tr}\left(\mathbf{W}^T\mathbf{W}\right)^j, \quad j = 1, 2, 3.$$

Under our PDB model, we have the following theoretical spectral moment,

$$\gamma_1 = \mu_1, \quad \gamma_2 = \mu_2 + c\mu_1^2, \quad \gamma_3 = \mu_3 + 3c\mu_1\mu_2 + c^2\mu_1^2$$

where

$$\mu_j = \hat{t}\hat{\sigma}_1^{2j} + (1-\hat{t})\hat{\sigma}_2^{2j}, \quad j = 1, 2, 3,$$

and $(\hat{t}, \hat{\sigma}_1, \hat{\sigma}_2)$ come from PDBLS algorithm.

**Theorem B.1.** *Under PDB model* (2) *with* $p/n = c$, *as* $n \to \infty$, *we have*

$$\hat{\gamma}_j - \gamma_j \xrightarrow{a.s.} 0 \quad j = 1, 2, 3. \tag{13}$$

## C. The difference between PUB and PDB model

In this section, we compare theoretical properties between our PDB model and the PUB model. Recall the theory on PDB model has been established in Theorems 3.1-3.3 and B.1. Next we list the corresponding results about PUB model(1).

The LSD of $\mathbf{W}^T\mathbf{W}$ under PUB model has density function (Martin & Mahoney, 2021; Staats et al., 2023) :

$$f(x; c, \sigma^2) = \frac{\sqrt{(\lambda_+ - x)(x - \lambda_-)}}{2\pi\sigma^2 cx} \mathbb{1}_{\{\lambda_- \leq x \leq \lambda_+\}}. \tag{14}$$

Here the boundary points $\lambda_{\pm}(\sigma^2) = \sigma_0^2(1 \pm \sqrt{c})^2$, $c = \frac{p}{n}$. And the sample spikes of $\mathbf{W}^T\mathbf{W}$ have the following relationship with population spikes $\alpha_j$ of $\mathbb{E}\mathbf{W}^T\mathbf{W}$ (Bai & Yao, 2012):

$$\lambda_j = \alpha_j + c\alpha_j \frac{\sigma^2}{\alpha_j - \sigma^2}, \quad j \in \{1, \dots, K\}. \tag{15}$$

Moreover, the relationship between the moments of $\mathbf{W}^T\mathbf{W}$ and $\mathbf{\Sigma}_{PUB}$, mainly determined by the bulk eigenvalues, is established in (Yao et al., 2015):

$$\frac{1}{p} \operatorname{tr} \left(\mathbf{W}^T\mathbf{W}\right)^j - \gamma_j \xrightarrow{a.s.} 0, \quad j = 1, 2, 3.$$

where

$$\gamma_1 = \mu_1, \quad \gamma_2 = \mu_2 + c\mu_1^2, \quad \gamma_3 = \mu_3 + 3c\mu_1\mu_2 + c^2\mu_1^2, \quad \mu_j = \hat{\sigma}^{2j}, \quad j = 1, 2, 3. \tag{16}$$

Here $\hat{\sigma}^{2j}$ can be obtained by BEMA (Ke et al., 2023) or Kernel (Staats et al., 2023) method.

The difference between the PUB and PDB model is summarized in Table 10.

*Table 10.* The difference between the PUB and PDB model.

|  | PUB model | PDB model |
| --- | --- | --- |
| LSD | (14) | (3)-(4) |
| spiked eigenvalues | (15) | (6)-(7) |
| spectral moment | (16) | (13) |

## D. Proof of Theorem 3.1

Firstly, we give some preliminary knowledge about Stieltjes transform, which plays a key role in our proofs. For any distribution function $G(x)$, its Stieltjes transform $m_G(z)$ is defined as

$$m_G(z) = \int \frac{1}{x - z} dG(x), \quad z \in \mathbb{C}^+,$$

where $\mathbb{C}^+ = \{z \in \mathbb{C} : \operatorname{Im}(z) > 0\}$ denotes the upper complex plane and $\operatorname{Im}(z)$ represents the imaginary part of $z$. Moreover $G(x)$ and $m_G(z)$ have a one-to-one relationship. The density function $G'(x)$ of $G(x)$ is given by Theorem 2.1 in (Silverstein & Choi, 1995):

$$G'(x) = \lim_{\eta \to 0} \frac{\operatorname{Im}(m_G(z))}{\pi}, \quad z = x + i\eta. \tag{17}$$

According to the main results in (Bai & Silverstein, 2010), the Stieltjes transform of LSD of $\mathbf{W}^T\mathbf{W}$ is $m(z)$, which satisfies the following equations:

$$z = -\frac{1}{\underline{m}(z)} + c\int \frac{v}{1+v\underline{m}(z)}dH(v), \tag{18}$$

$$\underline{m}(z) = -\frac{1-c}{z} + cm(z), \tag{19}$$

where $H(v)$ be the LSD of $\Sigma_{PDB}$. By the relation $\underline{m}(z) = -(1-c)/z + cm(z)$, along with Eq. (17), the density function of LSD of $\mathbf{W}^T\mathbf{W}$ is

$$\rho(x) = \lim_{\eta \to 0} \frac{\operatorname{Im} m(z)}{\pi} = \lim_{\eta \to 0} \frac{\operatorname{Im} \underline{m}(z)}{\pi c}, \quad z = x + i\eta, \quad \eta > 0,$$

from which we get Eq. (3).

And since the number of spiked eigenvalues $K$ is fixed, we have

$$
\begin{aligned}
H(v) &= \lim_{p\to\infty} \frac{1}{p}[\sum_{j=1}^{K} I(\alpha_j \le v) + (p-K)tI(\sigma_1^2 \le v) + (p-K)(1-t)I(\sigma_2^2 \le v)]\\
&= tI(\sigma_1^2 \le v) + (1-t)I(\sigma_2^2 \le v),
\end{aligned} \tag{20}
$$

which implies

$$
\begin{aligned}
z &= -\frac{1}{\underline{m}(z)} + c\int \frac{v}{1+v\underline{m}(z)}dH(v)\\
&= -\frac{1}{\underline{m}(z)} + ct\frac{\sigma_1^2}{1+\sigma_1^2\underline{m}(z)} + c(1-t)\frac{\sigma_2^2}{1+\sigma_2^2\underline{m}(z)}.
\end{aligned}
$$

Thus we obtain Eq. (4). The proof of Theorem 3.1 is complete.

## E. Proof of Theorem 3.2

By Eq. (4), this inverse function of $z \mapsto -1/\underline{m}(z)$ is

$$g(x) = x + cx\frac{t\sigma_1^2}{x-\sigma_1^2} + cx\frac{(1-t)\sigma_2^2}{x-\sigma_2^2}.$$

Here $g(x) = z$ and $x = -1/\underline{m}(z)$ in Eq. (4). Then according to Proposition 2.17 in (Yao et al., 2015) we obtain Theorem 3.2.

## F. Proof of Theorem 3.3

By Theorem 4.1 in (Bai & Yao, 2012) , the relationship between sample spikes $\lambda_j$ and population spikes $\alpha_j$ is established by

$$\lambda_j \xrightarrow{\text{a.s.}} \alpha_j + c\alpha_j \int \frac{v}{\alpha_j - v}dH(v), \quad j \in \{1,\ldots,K\}.$$

Moreover, according to Eq. (20), we have

$$\int \frac{v}{\alpha_j - v}dH(v) = \frac{t\sigma_1^2}{\alpha_j - \sigma_1^2} + \frac{(1-t)\sigma_2^2}{\alpha_j - \sigma_2^2},$$

from which for $j \in \{1,\ldots,K\}$, we obtain

$$\lambda_j \xrightarrow{\text{a.s.}} \alpha_j + c\alpha_j \frac{t\sigma_1^2}{\alpha_j - \sigma_1^2} + c\alpha_j \frac{(1-t)\sigma_2^2}{\alpha_j - \sigma_2^2} = g(\alpha_j)$$

Thus we complete the proof Theorem 3.3.

## G. Proof of Theorem 3.4

Recall the estimators

$$\widehat{\Theta}_{bulk} = \left\{\hat{\sigma}_1^2, \hat{\sigma}_2^2, \hat{t}\right\}, \ \widehat{\Theta}_{bound} = \left\{\hat{\lambda}_+, \hat{\beta}\right\}, \ \widehat{\Theta}_{spike} = \left\{\hat{K}, \hat{\alpha}_1, \ldots, \hat{\alpha}_{\hat{K}}\right\},$$

along with their corresponding population parameters

$$\Theta_{bulk} = \left\{\sigma_1^2, \sigma_2^2, t\right\}, \ \Theta_{bound} = \left\{\lambda_+, \beta\right\}, \ \Theta_{spike} = \left\{K, \alpha_1, \ldots, \alpha_K\right\}.$$

We now proceed to establish the consistency of these estimators.

For $\widehat{\Theta}_{bulk}$, its consistency relies on the convergence of the empirical spectral distribution (ESD)

$$F_n^{\mathbf{W}^T \mathbf{W}}(x) = \frac{1}{p} \sum_{j=1}^{p} \mathbb{1}_{\{\lambda_j \leq x\}}$$

to the limiting spectral distribution (LSD) characterized in Theorem 3.1. Based on the ESD, we obtain $\widehat{\Theta}_{bulk}$ via the procedures defined in Eq. (9)-(10). Moreover, Eq. (4) ensures a one-to-one correspondence between the population parameter $\Theta_{bulk} = \left\{\sigma_1^2, \sigma_2^2, t\right\}$ and LSD. Therefore, by Theorem 3.1, the ESD converges almost surely to the LSD, and applying Theorem 3.1 in (Li et al., 2013), we conclude that

$$\widehat{\Theta}_{bulk} \xrightarrow{\text{a.s.}} \Theta_{bulk}. \tag{21}$$

For $\widehat{\Theta}_{bound}$, the consistency of $\hat{\lambda}_+$ relies on the the convergence of $\widehat{\Theta}_{bulk}$ and the results in Theorem 3.2, while the consistency of $\hat{\beta}$ depends on the convergence of $\hat{K}$. Based on $\widehat{\Theta}_{bulk}$, we compute $\hat{\lambda}_+$ through Eq. (5)-(6), i.e.,

$$\hat{\lambda}_+ = \hat{g}(\hat{y}), \ \hat{y} = \arg\max_{x \in \mathbb{R}} \left\{\hat{g}'(x) = 0\right\}, \ \hat{g}(x) = x + c_n x \frac{\hat{t}\hat{\sigma}_1^2}{x - \hat{\sigma}_1^2} + c_n x \frac{(1 - \hat{t})\hat{\sigma}_2^2}{x - \hat{\sigma}_2^2}, \ c_n = p/n.$$

Moreover, Theorem 3.2 establishes a one-to-one correspondence between $\lambda_+$ and $\Theta_{bulk}$. Therefore, by Eq. (21) and the condition $c_n \to c$, we have $\hat{\lambda}_+ \xrightarrow{\text{a.s.}} \lambda_+$. Then by Eq. (11)-(12), we have $\hat{K} = \#\left\{\lambda_j \mid \lambda_j \in \left(\hat{\lambda}_+, \lambda_{\max}\right]\right\} \xrightarrow{\text{a.s.}} K$ and hence $\hat{\beta} = \lambda_{\hat{K}+p\hat{t}-\hat{K}\hat{t}} \xrightarrow{\text{a.s.}} \beta$. Thus we conclude that

$$\widehat{\Theta}_{bound} \xrightarrow{\text{a.s.}} \Theta_{bound}. \tag{22}$$

For $\widehat{\Theta}_{spike}$, its consistency relies on the convergence of $\widehat{\Theta}_{bulk}, \widehat{\Theta}_{bound}$ and the results in Theorem 3.3. By the convergence $\hat{\lambda}_+ \xrightarrow{\text{a.s.}} \lambda_+$ in $\widehat{\Theta}_{bound}$, we have $\hat{K} \xrightarrow{\text{a.s.}} K$. Given $\widehat{\Theta}_{bulk}$, according to Eq. (7), we compute $\hat{\alpha}_i$ by solving the equation

$$\hat{g}(\hat{\alpha}_j) = \lambda_j, \ \text{s.t.} \ \ \hat{g}'(\hat{\alpha}_j) > 0, \ \ j \in \{1, \ldots, \hat{K}\}.$$

Moreover, from Theorem 4.1 in (Bai & Yao, 2012) and Theorem 3.3, we have $\lambda_j \xrightarrow{\text{a.s.}} g(\alpha_j), \ g'(\alpha_j) > 0, \ j \in \{1, \ldots, K\}$. Therefore by Eq. (21), $\hat{K} \xrightarrow{\text{a.s.}} K$ and $c_n \to c$, we obtain

$$\widehat{\Theta}_{spike} \xrightarrow{\text{a.s.}} \Theta_{spike}. \tag{23}$$

Finally, by Eq. (21), (22), (23), we complete the proof of Theorem 3.4.

## H. Proof of Theorem B.1

According to Lemma 2.16 in (Yao et al., 2015), we have the following relationship:

$$\gamma_1 = \mu_1, \ \gamma_2 = \mu_2 + c\mu_1^2, \ \gamma_3 = \mu_3 + 3c\mu_1\mu_2 + c^2\mu_1^2,$$

where

$$\gamma_j = \lim_{p\to\infty} \frac{1}{p} \operatorname{tr}\left(\mathbf{W}^T\mathbf{W}\right)^j, \quad \mu_j = \int v^j dH(v), \quad j = 1, 2, 3.$$

Moreover, together with Eq. (20), we have

$$\mu_j = t\sigma_1^{2j} + (1-t)\sigma_2^{2j}, \quad j = 1, 2, 3.$$

And by the definition of $\mathbf{\Sigma}_{PDB}$,

$$\lim_{p\to\infty} \frac{1}{p} \operatorname{tr}(\mathbf{\Sigma}_{PDB})^j = t\sigma_1^{2j} + (1-t)\sigma_2^{2j}, \quad j = 1, 2, 3,$$

we obtain (13). Thus we complete the proof Theorem B.1.

