# OpenReview forum: "Weight matrices compression based on PDB model in deep neural networks"
_ICML.cc/2025/Conference — ICML 2025 poster_

### Official Review · Reviewer_LPdE · 2025-03-10

**Overall Recommendation:** 3

**Summary:**

This paper studies the problem of DNN weight compression for seek of better generalization. They propose the Population Double Bulk (PDB) model to characterize the eigenvalue behavior of $\mathbf{W}^\top\mathbf{W}$ in the bulk+spikes phase during DNN training, generalizing the existing Population Unit Bulk (PUB) model to allow for finer characterization of the bulk eigenvalues (from one bulk in PUB to two bulks in PDB). The paper further investigates the asymptotic limits of the proposed model based on Random Matrix Theory (RMT), which allows for new algorithm design to (i) estimate the PDB model parameters based on observed empirical eigenvalues; and (ii) compress the model based on the estimated PDB model by pruning the eigenvalues inside the smaller bulk (second bulk). Empirical results demonstrate the effectiveness of the proposed method in terms of improving generalization.

**Claims And Evidence:**

Mostly yes.

**Essential References Not Discussed:**

To the knowledge of the reviewer there are no lacking essential references.

**Experimental Designs Or Analyses:**

Yes.

**Methods And Evaluation Criteria:**

Yes.

**Other Comments Or Suggestions:**

Please see the **Weakness** part of the review.

**Other Strengths And Weaknesses:**

**Strengths:** From the reviewer's perspective, the most important contribution of the paper is that the proposed new model of the weight matrix LSD (PDB) gives better approximation of the empirical spectral distribution given the experimental results shown in the paper. This better approximation also results in potentially improved compression algorithm in terms of generalization performance. The model is clean and quite direct to obtain from existing models.

**Weakness:**
1. The improvement of the testing accuracies w.r.t. the base model on the tasks considered in the paper is relatively limited (mostly less than 1%), which makes the strength of the overall method weakened.
2. The paper only discusses PUB model and the compression methods based PUB. Little is discussed and compared about other potential compression based method for improving generalization.
3. What is the computational overhead of the compression algorithm proposed in this paper compared with the algorithms mentioned in the paper that are based on PUB? The paper does not touch on this perspective.

**Questions For Authors:**

Please see the **Weakness** part of the review.

**Relation To Broader Scientific Literature:**

The paper contributes to the literature of DNN weight compression based on the model of population bulks. Specifically, the paper proposes a finer characterization of the limiting spectral distribution via including two classes of bulked eigenvalues. New algorithms to compress the weight matrices are proposed accordingly.

**Theoretical Claims:**

Yes. But the reviewer is concerned with the proof of Theorem 3.4 (Appendix G). The proofs are overly simplified and should be more concrete to make the paper self content and convincing.

---

> ### Author Rebuttal · Authors · 2025-04-01
>
> Thanks for your recognition and the valuable suggestions. Please find our response below.
>
> (**T** for Theoretical Claims, **W** for Weakness)
>
> **1. Detailed proof of Theorem 3.4  [T]**
>
> We are sorry for the overly simplified proof of Theorem 3.4 in our paper.  We will include more detailed and concrete proofs of Theorem 3.4 in the final version. Below is a brief outline.
>
> 1. For $\widehat\Theta_\text{bulk}$ , its consistency relies on the convergence of ESD $F_n^{W^TW}(x)=\frac1p\sum_\text{j=1}^p \mathbb{1}(\lambda_j \leq x)$ to the LSD (defined on page 1 in our paper). Specifically,
>
> $$
> \widehat\Theta_\text{bulk} \xLeftrightarrow{Eq. (9)(10)} ESD\xrightarrow{\text{Theorem 3.1}} LSD \xLeftrightarrow{Eq. (4)} \Theta_\text{bulk}
> $$
>
> 2. For $\widehat\Theta_\text{bound}$, its consistency relies on the convergence of $\widehat{\Theta}_\text{bulk}$. Specifically,
>
> $$
> \widehat\Theta_\text{bound} \xLeftrightarrow{Eq. (5)(11)(12)} \widehat\Theta_\text{bulk} \xrightarrow{\text{Theorem 3.1}} \Theta_\text{bulk} \xLeftrightarrow[Eq.(5)(6)]{\text{Theorem 3.2}} \Theta_\text{bound}
> $$
>
> 3. For $\widehat\Theta_\text{spike}$, its consistency relies on the convergence of $\widehat\Theta_\text{bulk}$, $\widehat\Theta_\text{bound}$ and $\lambda_\text{spike}$. Specifically,
>
> $$
> \widehat\Theta_\text{spike}\xLeftrightarrow{Eq. (6)(7)(11)}\widehat\Theta_\text{bulk}, \widehat\Theta_\text{bound},\lambda_\text{spike} \xrightarrow{\text{Theorems 3.1-3.3}}\Theta_\text{bulk},\Theta_\text{bound},g(\alpha_\text{spike})\xLeftrightarrow{Eq.(7)(11)}\Theta_\text{spike}
> $$
>
> **2.About improvement of test accuracy  and comparison with additional methods [W1,W2]**
>
> In model compression, besides accuracy, another important metric is the **compression ratio** (ratio of rank of $W$ after and before compression). We aim to compress $W$ while maintaining accuracy, seeking **a balance between accuracy and compression ratio**, rather than merely pursuing high accuracy. In this rebuttal, we include **additional experiments on BERT and T5-base** and compare with **two other compression methods** :
>
> ​    1. navie SVD (using 0.55 as the empirical compression ratio, Shmalo et al. (2023)),
>
> ​    2. sparse low-rank (SLR) (Sridhar Swaminathan et al., *Neurocomputing*, 2020).
>
> The table below presents the accuracy and compression ratio.
>
> |     Network      | Noise | Base  |       PDB        |     SLR      |     PUB      | naive SVD  |
> | :--------------: | :---: | :---: | :--------------: | :----------: | :----------: | :--------: |
> |    FCNN:MNIST    |  30%  | 0.801 |   0.808(30.1%)   | 0.789(27.3%) | 0.793(13.9%) |   0.803    |
> | ResNet18:CIFAR10 |  30%  | 0.642 |   0.658(15.8%)   | 0.619(11.1%) | 0.64 (7.2%)  |   0.647    |
> |  VGG16:CIFAR10   |  30%  | 0.699 |   0.705(6.4%)    | 0.701(9.1%)  | 0.686 (4.3%) |    0.7     |
> |    BERT: RTE     |  0%   | 0.703 |   0.732(2.3%)    | 0.717(3.3%)  | 0.717 (9.6%) |   0.703    |
> |  BERT: SCITAIL   |  0%   | 0.906 |   0.916(2.1%)    | 0.918(3.3%)  |  0.913(12%)  |   0.906    |
> |   T5-base: RTE   |  0%   | 0.717 | **0.754(1.7%)**  | 0.725(16.3%) | 0.732(8.1%)  |   0.717    |
> | T5-base: SCITAIL |  0%   | 0.921 |   0.924(2.3%)    | 0.901(9.8%)  | 0.917(4.7%)  |    0.92    |
> |     Average      |       | 0.77  | **0.785(8.67%)** | 0.767(11.5%) | 0.771(8.5%)  | 0.771(55%) |
>
> Our method achieves the best accuracy, with a **maximum improvement of 5.2%** ($\frac{0.754}{0.717}-1$), and maintains a competitive compression rate, reaching 1.7%  for T5-base: RTE.
>
>  Moreover, in the model compression literature, accuracy improvements in general are not substantial, e.g., 0.7% in Hyeji Kim et al. (CVPR, 2019) and 0.36%–0.55% in Georgios Georgiadis (CVPR, 2019).
>
> **3. Computational overhead of compression algorithm [W3]**
>
> The table below compares the computational efficiency, with time measured in seconds for total execution and inference after data processing on NVIDIA L40 GPUs (Ubuntu 22.04).
>
> |   DNN   |   PDB    | PUB  | naive SVD | SLR  |
> | :-----: | :------: | :--: | :-------: | :--: |
> |  VGG16  |    42    |  41  |    47     |  53  |
> | Resnet  |    19    |  15  |    18     |  23  |
> |   VIT   |   236    | 231  |    243    | 256  |
> | T5-base |    47    |  44  |    45     |  42  |
> |  Bert   |    20    |  15  |    18     |  29  |
> | Average | **72.8** | 69.2 |   74.2    | 80.6 |
>
>  As shown, the computational overhead of our PDB model is slightly higher than PUB. However, this additional overhead is relatively modest and is outweighed by the improvements in compression performance and model accuracy.

---

> > ### Comment · Reviewer_LPdE · 2025-04-04
> >
> > Thanks for the detailed response, and most of my concerns are addressed. I have raised my score to 3 accordingly.

---

> > > ### Author Response · Authors · 2025-04-04
> > >
> > > We sincerely appreciate your valuable comment and the time you dedicated to reviewing our work. Thank you for recognizing our revisions and  kind support.

---

### Official Review · Reviewer_9rcu · 2025-03-13

**Overall Recommendation:** 4

**Summary:**

The paper introduces the population double bulk (PDB) model, an extension of the population unit bulk (PUB) model, to provide a more accurate description of the spectral properties of the weights in deep neural networks. In the PDB model, the informative components of the spectrum are captured by the spikes and the first bulk, while the second bulk corresponds to noise. Empirical results demonstrate that the PDB model outperforms the PUB model in terms of both performance and alignment. Furthermore, the PDB model is used to compress the weight matrices of convolutional neural networks (CNNs) into a low-rank structure by discarding the singular values associated with the second bulk. Networks compressed using the PDB model achieve superior accuracy compared to those compressed with the PUB model.

**Claims And Evidence:**

Tables 3-4 and Figure 3 demonstrate that the PDB model captures the empirical observations more effectively than the PUB model.

The paper claims that the proposed algorithms offer the optimal compression ratio. While Figure 6 provides some support for this claim, there is a concern regarding whether the ratio is indeed the best recommendation. Specifically, the performance does not show a significant decline immediately after the values of $\beta$ or $\lambda_+$ in Figure 6.

**Essential References Not Discussed:**

Related works are well addressed.

**Experimental Designs Or Analyses:**

The experimental setup for the DNN training is consistent with established practices in the literature. However, it remains unclear whether the PDBLS and noise-filtering algorithms offer a better compression ratio than the PUB models. Figure 6 presents only the parameters of the PDB models, and Table 5 displays only the compression results for the PDB model, making it difficult to directly compare the compression quality between PDB and PUB models.

Additionally, it would be insightful to include a comparison between the PDB model and a naïve low-rank compression method that selects the top-K singular values.

**Methods And Evaluation Criteria:**

The proposed methods and evaluation criteria are appropriately chosen for the problem.

**Other Comments Or Suggestions:**

The text in the figures is too small. It is recommended to increase the font size in the figures for better readability.

**Other Strengths And Weaknesses:**

**Strengths**

1. The paper is well-motivated by the fact that the PUB model is based on restrictive assumptions. The proposed spectral model effectively addresses this issue, enhancing the practicality of the population bulk model family.
2. The proposed method has significant potential for adoption across various subcategories of machine learning and deep learning.

**Weaknesses**

Most of my concerns relate to clarity.

1. It is unclear under what assumptions the training process produces the bulk+spike phase.
2. The experimental results could be further strengthened, as discussed in the "Experimental Designs or Analyses" section. Specifically, the proposed matrix compression scheme should be compared with both the PUB model and the naïve top-K singular value selection method.
3. Although the double bulk model resembles the empirical spectral distribution, it is unclear why the first bulk is considered the information criterion, rather than the noise criterion.

**Questions For Authors:**

1. How much improvement in compression quality (e.g., accuracy change vs. compression ratio) does the proposed PDBLS + noise-filtering algorithm offer compared to the PUB model or naïve low-rank compression methods?

2. What are the key motivations for choosing the PDB model over the naïve low-rank compression methods?

3. Could the authors provide insights or a theoretical discussion on why $\beta$ is chosen as the information-noise boundary? Does the double-bulk distribution emerge naturally during the training process?

**Relation To Broader Scientific Literature:**

The algorithm to identify the signal and the noise in the spectral density would be useful in many fields of machine learning and numerical methods for scientific applications.

**Theoretical Claims:**

I checked the correctness of the proof of Theorem 3.1 and it seems correct.

---

> ### Author Rebuttal · Authors · 2025-04-01
>
> Thanks for your recognition and the valuable suggestions. Please find our response below.
>
> (**C** for Claims And Evidence, **E** for Experimental Designs,  **W** for weakness, **Q** for Questions)
>
> **1. Whether the ratio is the best [C]**
>
> Our primary goal is to identify the noise-information boundary and remove the noisy eigenvalues. Once the signal eigenvalues are removed, accuracy begins to degrade. Therefore, we aim to locate the point where **accuracy first starts to decline**, rather than the point of significant decline. In Figure 6, a **significant drop in accuracy** indicates that the model is **over-compressed**. The boundary point $\beta$ from PDB better captures the **initial decline points** compared to $\lambda_{+}$from  PUB, showing that PDB better aligns with the behavior of eigenvalues of $W$.
>
> Moreover, most $\beta$ in Figure 6 are located near the initial decline points, though a few cases are less obvious. This is because $\beta$ is an asymptotic limit based on PDB when the dimension of $W$ tends to infinity. There exists a gap between the asymptotic limit and the finite-sample empirical value. Additionally, the random error of  PDB estimate can also contribute to this effect.
>
> **2. Additional experiments [E,W2,Q1]**
>
> We are sorry for the lack of a detailed comparison and clear explanation in the main text. We include **additional experiments on BERT and T5-base** and compare with **two other methods** :
>
> ​    1. naive SVD (using 0.55 as the empirical compression ratio, Shmalo et al. (2023)),
>
> ​    2. sparse low-rank (SLR) (Sridhar Swaminathan et al., *Neurocomputing*, 2020).
>
> The table below presents the accuracy and compression ratio (ratio of rank of $W$ after and before compression).
>
> |     Network      | Noise | Base  |       PDB        |     SLR      |     PUB      | naive SVD  |
> | :--------------: | :---: | :---: | :--------------: | :----------: | :----------: | :--------: |
> |    FCNN:MNIST    |  30%  | 0.801 |   0.808(30.1%)   | 0.789(27.3%) | 0.793(13.9%) |   0.803    |
> | ResNet18:CIFAR10 |  30%  | 0.642 |   0.658(15.8%)   | 0.619(11.1%) | 0.64 (7.2%)  |   0.647    |
> |  VGG16:CIFAR10   |  30%  | 0.699 |   0.705(6.4%)    | 0.701(9.1%)  | 0.686 (4.3%) |    0.7     |
> |    BERT: RTE     |  0%   | 0.703 |   0.732(2.3%)    | 0.717(3.3%)  | 0.717 (9.6%) |   0.703    |
> |  BERT: SCITAIL   |  0%   | 0.906 |   0.916(2.1%)    | 0.918(3.3%)  |  0.913(12%)  |   0.906    |
> |   T5-base: RTE   |  0%   | 0.717 | **0.754(1.7%)**  | 0.725(16.3%) | 0.732(8.1%)  |   0.717    |
> | T5-base: SCITAIL |  0%   | 0.921 |   0.924(2.3%)    | 0.901(9.8%)  | 0.917(4.7%)  |    0.92    |
> |     Average      |       | 0.77  | **0.785(8.67%)** | 0.767(11.5%) | 0.771(8.5%)  | 0.771(55%) |
>
> Our method achieves the best accuracy. Moreover, Figure 6 presents the parameters for both PDB and PUB. The vertical dashed lines represent the suggested optimal compresion ratio, $\beta$ for PDB and  $\lambda_+$ for PUB.
>
> **3. Assumptions to produce the bulk+spike phase [W1]**
>
> Martin & Mahone (JMLR, 2021) point out that the bulk+spike phase typically emerges when $W$ exhibit **weak self-regularization** during training. To the best of our knowledge, so far there is no rigorous mathematical hypothesis specifying when the bulk+spike phase occurs. It is more of an **empirically observed phenomenon**. Both PUB and PDB provide mathematical frameworks for modeling this phenomenon.
>
> **4. About noise-information boundary $\beta$ [W3,Q3]**
>
> The entries of the **initial** $W_0$  are pure noise, and $EW_0^TW_0=\sigma_0^2I_p$. The eigenvalues of $W_0^TW_0$ follow  a  unimodal MP law which contains **no information**. As training progresses, the weight matrix is updated with new gradients, which alters the structure of  $EW^TW$.  Martin \& Mahoney (JMLR, 2021) conducted extensive analyses on the eigenvalue distribution of $W^TW$ during training. They claim that, as training proceeds, the bulk distribution becomes **bimodal** and **large eigenvalues emerge**, forming the **bulk+spike phase**, see Figures 11,18 in their paper.  We provide a theoretical explanation for this phenomenon,  which is PDB for $EW^TW$. Correspondingly, we treat the **emerging additional bulk and spikes** as **information**, while the **smaller bulk** is still **noise**.  $\beta$  separates the first and second bulk, thus it's chosen as the noise-information boundary.
>
> **5. Key motivations for choosing  PDB [Q2]**
>
> Both PDB and PUB are low-rank compression methods. The key challenge lies in choosing the optimal compression ratio. We first provide a mathematical framework PDB for the eigenvalues in the bulk+spike phase. Then the compression ratio is determined based on their own eigenvalue behavior. Additionally, PDB debiases large eigenvalues to enhance robustness. It can improve accuracy with noisy labels. In contrast, naive methods rely on **empirical heuristics** (same ratio for all matrices) and don't correct for signal eigenvalues, lacking theoretical support.

---

> > ### Comment · Reviewer_9rcu · 2025-04-04
> >
> > Thank you for the detailed response and the additional experimental results. My main concern was the lack of comparison with SVD-based methods, which has been adequately addressed in the authors' response. I have raised my score to a 4, as I believe PDB makes a meaningful contribution to the problem of layer-wise adaptive low-rank compression.

---

> > > ### Author Response · Authors · 2025-04-05
> > >
> > > We sincerely appreciate your kind support and recognition of our work. We are delighted that our response and additional experiments have addressed your concerns, and we are truly grateful for your valuable feedback.

---

### Official Review · Reviewer_4M7a · 2025-03-13

**Overall Recommendation:** 3

**Summary:**

The paper proposed a population double bulk model for compressing weight matrices in neural networks. Compared to previous pupulation unit bulk model, the PDB model has more parameters and better approximates the weight matrices. Theoretical analysis (drawing tools from random matrix theory) and algorithms are proposed to estimate the parameters in PDB model. Experiments on neural networks shows an accuracy improvement using PDB model compared to using PUB model.

**Claims And Evidence:**

Yes.

**Essential References Not Discussed:**

N/A

**Experimental Designs Or Analyses:**

Looks reasonable to me.

**Methods And Evaluation Criteria:**

Makes sense to me.

**Other Comments Or Suggestions:**

N/A

**Other Strengths And Weaknesses:**

Strengths: The paper provides solid works including proposing an extended model of PUB, providing corresponding theoretical analysis and algorithms to estimate the parameters, and running experiments on neural networks that show improvement.

Weaknesses: These are not necessarily weaknesses but more of questions.

1. Is it assumed that the weight matrices are i.i.d.? How accurately does the PUB/PDB model approximate the real weight matrices. Were comparisons between PUB and data-driven methods made in existing literatures? Is it possible to comment on how well PUB/PDB models perform compared to data-driven methods?

2. While it is mentioned that matrix compression is set out to mitigate the overfitting, it seems that no compression performs almost equally well in the experiments and better when the dataset is good enough.

3. Is the matrix compression performed for each layer or specific layers?

4. It was mentioned that in the training process the third phase, the flat-tail phase is less common. Is it the case that most models stop at the spike and bulk phase during the training proccess?

**Questions For Authors:**

See above.

**Relation To Broader Scientific Literature:**

The paper discussed the problem of reducing overfitting through the compression of weight matrices, which seems to be of interest.

**Theoretical Claims:**

Looks correct to me.

---

> ### Author Rebuttal · Authors · 2025-04-01
>
> Thanks for your recognition and the valuable suggestions. Please find our response below. (**W** for weakness)
>
> **1. Model details [W1]**
>
> The entries of the **initial** weight matrix  $W_0$ are **i.i.d.** with mean 0 and variance $\sigma^2$, resulting in $\mathbb{E}W_0^TW_0=\sigma_0^2 I_p$. As **training progresses**, PDB assumes that $\mathbb{E}W^TW=\operatorname{diag}\left(\alpha_1, \ldots, \alpha_K, \sigma_1^2, \ldots, \sigma_1^2, \sigma_2^2, \ldots, \sigma_2^2\right)$ , indicating that the entries are **no longer i.i.d.** due to  heterogeneous variances.
>
> We use  the density curve (Figure 3) and spectral moments of $W$ (Tables 3-4) to assess model approximation performance. We also compare the first three theoretical spectral moments with the empirical values on **additional neural networks**.
>
> |           |  |     T5-base: RTE  |       | |      Bert: SCITAIL  |       |
> | :-------: | :----------: | :---: | :---: | :-----------: | :---: | :---: |
> |  Moments  |    $\gamma_1$     | $\gamma_2$ | $\gamma_3$ |     $\gamma_1$     | $\gamma_2$ | $\gamma_3$ |
> |    PUB    |     0.67     |  0.9  | 1.51  |     0.59      | 0.69  | 1.02  |
> |    PDB    |     0.77     | 1.55  | 4.17  |     0.67      | 1.18  | 2.78  |
> | empirical |     0.72     | 1.83  | 5.35  |     0.71      | 1.38  | 3.95  |
>
> Actually, the method proposed by Staats et al. (Physical Review E, 2023) is **data-driven** and  we provide a  detailed comparison in Table 6 of our paper Appendix.  We also include **additional experiments on BERT and T5-base** and compare with **two other compression methods** :
>
>  1.naive SVD (using 0.55 as the empirical compression ratio, Shmalo et al. (2023)),
>
>  2.sparse low-rank (SLR) (Sridhar Swaminathan et al., *Neurocomputing*, 2020).
>
> The table  below shows the accuracy and compression ratio (ratio of rank of $W$ after and before compression) .
>
> |     Network      | Noise | Base  |       PDB        |     SLR      |     PUB      | naive SVD  |
> | :--------------: | :---: | :---: | :--------------: | :----------: | :----------: | :--------: |
> |    FCNN:MNIST    |  30%  | 0.801 |   0.808(30.1%)   | 0.789(27.3%) | 0.793(13.9%) |   0.803    |
> | ResNet18:CIFAR10 |  30%  | 0.642 |   0.658(15.8%)   | 0.619(11.1%) | 0.64 (7.2%)  |   0.647    |
> |  VGG16:CIFAR10   |  30%  | 0.699 |   0.705(6.4%)    | 0.701(9.1%)  | 0.686 (4.3%) |    0.7     |
> |    BERT: RTE     |  0%   | 0.703 |   0.732(2.3%)    | 0.717(3.3%)  | 0.717 (9.6%) |   0.703    |
> |  BERT: SCITAIL   |  0%   | 0.906 |   0.916(2.1%)    | 0.918(3.3%)  |  0.913(12%)  |   0.906    |
> |   T5-base: RTE   |  0%   | 0.717 | **0.754(1.7%)**  | 0.725(16.3%) | 0.732(8.1%)  |   0.717    |
> | T5-base: SCITAIL |  0%   | 0.921 |   0.924(2.3%)    | 0.901(9.8%)  | 0.917(4.7%)  |    0.92    |
> |     Average      |       | 0.77  | **0.785(8.67%)** | 0.767(11.5%) | 0.771(8.5%)  | 0.771(55%) |
>
> It can be seen that while the compression raio of PDB lies between PUB and SLR, it achieves the best accuracy, with a **maximum improvement of 5.2%** ($\frac{0.754}{0.717}-1$).
>
> **2. About overfitting [W2]**
>
> Our primary goal is to **reduce model complexity** while **preserving generalization ability**. During the matrix compression process, some model information is inevitably lost. Our approach aims to **identify the optimal boundary between noise and information**, effectively removing **noisy eigenvalues** to maintain the model's generalization without excessive information loss. When the data is clean, the impact of noisy eigenvalues is minimal, making the difference between compressed and uncompressed matrices negligible. However, when the data quality is poor, the noise embedded in $W$ can negatively affect test accuracy. Since our compression method essentially **filters out noise**, it **enhances generalization** and improves the model’s robustness. Therefore, our method shows a more significant advantage in the presence of noise, whereas when the data quality is good, the difference between compressed and uncompressed models is minimal.
>
> **3. Is the matrix compression performed for each layer or specific layers? [W3]**
>
> We **selectively compress specific layer matrices**, particularly those where the size of $W$ is sufficiently large. For smaller matrices, the **limited number of eigenvalues** results in **less accurate model estimation**, making compression ineffective.
>
> **4. About spike and bulk phase [W4]**
>
> Martin & Mahone (JMLR, 2021) conducted a **comprehensive and in-depth analysis** of the eigenvalue distribution of $W^TW$ throughout the training process. They found that when the DNN weight matrices exhibit **weak self-regularization**，**eigenvalue mass shifts to larger values**, forming the Bulk+Spikes phase during the first few epochs . Once the spikes appear, **substantial changes** in the distribution become hard to see.

---

### Official Review · Reviewer_nNCM · 2025-03-14

**Overall Recommendation:** 3

**Summary:**

This paper presents the Population Double Bulk (PDB) model, an extension of the Population Unit Bulk (PUB) model, for the effective compression of deep neural network weight matrices. The authors leverage a dual-cluster structure to more accurately analyze the eigenvalue distribution and employ the PDBLS algorithm to efficiently estimate the boundary between informative signals and noise. Additionally, they introduce the PDB Noise-Filtering algorithm, which selectively removes unnecessary eigenvalues while preserving crucial information, thereby improving compression performance.

The experimental results demonstrate that the proposed PDB-based compression technique can maintain the same test accuracy as existing methods at lower ranks or even enhance generalization performance. Furthermore, the approach exhibits robustness in noisy data scenarios, suggesting its practical applicability in real-world deep learning tasks.

## update after rebuttal
My main concern was the lack of evaluation on other models such as LLMs and the comparison with SVD-based methods, which was well addressed in the authors’ reply. For this reason, I have raised my score from 2 to 3.

**Claims And Evidence:**

- The paper conducts LSD (eigenvalue distribution) comparison experiments, asserting that the PDB model provides a more accurate representation of the weight matrices in real neural networks than the PUB model.
- The study suggests that the PDB model can enhance generalization performance by regulating the Lipschitz constant.
- While the paper claims that PDB-based compression outperforms the PUB-based approach, a more comprehensive comparative analysis with competing methods is necessary to elucidate the reasons behind its superior performance.

**Essential References Not Discussed:**

- None

**Experimental Designs Or Analyses:**

- The paper evaluates the PDB model against the PUB model using a range of neural networks (FNN, ResNet18, VGG16) and benchmark datasets (MNIST, CIFAR-10, ImageNet).
- To assess generalization, experiments were also conducted on noisy data.
- However, the study focuses solely on CNN-based architectures, leaving open the question of whether the PDB model would perform similarly well in Transformer-based models like BERT or ViT.
- Additionally, the method has only been tested on computer vision tasks, and its effectiveness in other domains, such as NLP or speech recognition, remains unexplored.

**Methods And Evaluation Criteria:**

- To address the limitations of the existing PUB model, the authors introduce the Population Double Bulk (PDB) model and develop the PDBLS algorithm and PDB Noise-Filtering algorithm based on it.
- Notably, the proposed approach effectively analyzes the eigenvalue distribution of neural networks and applies spectral analysis using Random Matrix Theory (RMT) to provide theoretical justification for the compression method.
- However, since the evaluation is limited to a comparison with the PUB model, the study lacks a broader assessment against state-of-the-art compression techniques.
- Additionally, it remains unclear whether the proposed method is optimized specifically for image data or if it can be generalized to other modalities, such as speech or text data. Further experiments on diverse datasets would strengthen the claims regarding the method’s applicability.

**Other Comments Or Suggestions:**

- None

**Other Strengths And Weaknesses:**

Strengths
- This paper expands the spectral analysis of neural network weight matrices by introducing the PDB model, which provides a more accurate explanation of eigenvalue distributions compared to the existing PUB model.
- By applying Random Matrix Theory (RMT), the study establishes a theoretical foundation for understanding eigenvalue behavior and introduces the PDBLS algorithm, which enables a more precise separation between informative signals and noise.
- The effectiveness of the proposed approach is demonstrated through experiments on a variety of neural network architectures (FNN, ResNet18, VGG16) and benchmark datasets (MNIST, CIFAR-10, ImageNet), highlighting its practical applicability.
- The PDB model shows strong generalization performance, even in the presence of noisy data, and in some cases, it improves over existing methods, suggesting its potential to help mitigate overfitting.

Weaknesses
- The study does not include a direct comparison with other well-established neural network compression techniques (e.g., SVD), making it difficult to determine whether the PDB-based approach offers a clear advantage.
- The experiments are limited to CNN-based architectures, leaving it unclear how well the PDB model performs on other structures, such as Transformers (BERT, ViT).
- The method has only been tested on computer vision tasks, so its applicability to other domains, such as natural language processing (NLP) and speech recognition, remains uncertain.
- (Optional) The paper does not discuss the computational efficiency of the compressed models (FLOPs reduction, inference speed, memory savings), making it hard to assess their practical benefits.

**Questions For Authors:**

- Can the authors provide experiments on how well the PDB model performs on other architectures such as Transformers (BERT, ViT)?
- Could the authors provide experiments on application to other domains such as Natural Language Processing (NLP)?

**Relation To Broader Scientific Literature:**

- Previous research has used the Population Unit Bulk (PUB) model to analyze the eigenvalue distribution of weight matrices, but its single-variance assumption fails to fully capture the complexity of real neural networks.
- To address this, the paper introduces the Population Double Bulk (PDB) model, which considers two bulk distributions, allowing for a more realistic representation of eigenvalue behavior.
- However, further comparative experiments with existing compression methods are needed to better assess the practical performance of the PDB model.

**Theoretical Claims:**

- While the PUB model assumes a single variance (σ²), the PDB model extends this assumption by incorporating two bulk distributions, claiming to better model general cases. However, additional experimental and theoretical validation is required to determine whether the number of bulks is inherently limited to two in real-world data.
- The study leverages Random Matrix Theory (RMT) to demonstrate that the eigenvalue distribution in the PDB model follows a specific equation, ensuring theoretical consistency. However, further empirical validation on diverse neural network architectures is necessary to confirm the practical applicability of the proposed approach.

---

> ### Author Rebuttal · Authors · 2025-04-01
>
> Thanks for your recognition and the valuable suggestions. Please find our response below.(**T** for Theoretical Claims, **W** for weakness, **Q** for Questions)
>
> **1. Comparison with additional methods [W1]**
>
> We include additional experiments on two other methods:
>
> ​    1. naive SVD (using 0.55 as the empirical compression ratio, Shmalo et al. (2023)),
>
> ​    2. sparse low-rank (SLR) (Sridhar Swaminathan et al., *Neurocomputing*, 2020).
>
> Table below shows accuracy and compression ratio (ratio of rank of $W$ after and before compression).
>
> | Network          | Noise | Base  |       PDB        |      PUB      |  Naive SVD  |      SLR      |
> | ---------------- | :---: | :---: | :--------------: | :-----------: | :---------: | :-----------: |
> | FCNN:MNIST       |  0%   | 0.98  |   0.98 (15.7%)   | 0.979  (8.6%) |    0.98     | 0.98 (15.6%)  |
> |                  |  30%  | 0.801 |  0.808 (30.1%)   | 0.793 (13.9%) |    0.803    | 0.789 (27.3%) |
> | ResNet18:CIFAR10 |  0%   | 0.835 |  0.838 (21.5%)   | 0.834 (10.4%) |    0.835    |  0.836 (26%)  |
> |                  |  30%  | 0.642 |  0.658 (15.8%)   | 0.64  (7.2%)  |    0.647    | 0.619 (11.1%) |
> | VGG16:CIFAR10    |  0%   | 0.842 |  0.842 (13.5%)   | 0.841  (5.5%) |    0.842    |  0.842 (13%)  |
> |                  |  30%  | 0.699 |  0.705  (6.4%)   | 0.686  (4.3%) |     0.7     | 0.701  (9.1%) |
> | **Average**      |       | 0.800 | **0.805(17.2%)** | 0.795  (8.3%) | 0.801 (55%) | 0.794 (17.2%) |
>
> **2. Additional experiments on LLM [W2,W3,Q1,Q2]**
>
> We include additional experiments on BERT, ViT, and T5-base. Table below presents accuracy and compression ratio. SLR performs poorly on VIT(accuracy 0.1), thus omitted.
>
> |     Network      | Base  |       PDB        | SLR           | PUB          | naive SVD   |
> | :--------------: | ----- | :--------------: | ------------- | ------------ | ----------- |
> |    BERT: RTE     | 0.703 |   0.732 (2.3%)   | 0.717 (3.3%)  | 0.717 (9.6%) | 0.703       |
> |  BERT: SCITAIL   | 0.906 |   0.916 (2.1%)   | 0.918 (3.3%)  | 0.913 (12%)  | 0.906       |
> |   T5-base: RTE   | 0.717 |   0.754 (1.7%)   | 0.725 (16.3%) | 0.732 (8.1%) | 0.717       |
> | T5-base: SCITAIL | 0.921 |   0.924 (2.3%)   | 0.901 (9.8%)  | 0.917 (4.7%) | 0.92        |
> |    VIT-L: DTD    | 0.745 |   0.753 (11%)    | -             | 0.748 (8.2%) | 0.741       |
> |  VIT-L: SUN397   | 0.768 |  0.777 (13.4%)   | -             | 0.772 (11%)  | 0.772       |
> |   **Average**    | 0.793 | **0.809 (5.5%)** | -             | 0.800 (8.9%) | 0.793 (55%) |
>
> **3.Computational efficiency [W4]**
>
> The table below compares the computational efficiency, with time measured in seconds for total execution and inference after data processing on NVIDIA L40 GPUs (Ubuntu 22.04).
>
> |   DNN   |   PDB    | PUB  | naive SVD | SLR  |
> | :-----: | :------: | :--: | :-------: | :--: |
> |  VGG16  |    42    |  41  |    47     |  53  |
> | Resnet  |    19    |  15  |    18     |  23  |
> |   VIT   |   236    | 231  |    243    | 256  |
> | T5-base |    47    |  44  |    45     |  42  |
> |  Bert   |    20    |  15  |    18     |  29  |
> | Average | **72.8** | 69.2 |   74.2    | 80.6 |
>
> **4. Validity of PDB[T1]**
>
> **On the Empirical Side**: Our  motivation is that the spectral distribution of $W^TW$ does not perfectly align with PUB. This led us to consider more general models. Since the size of $W$ is typically too small to guarantee an accurate estimation of a continuous population bulk, we opted for a discrete surrogate of a M-bulk model ($M\geq2$). However, through extensive experiments (Section 5 included), we found that the proportions of  **bulks beyond M=2** are nearly negligible (Table 1). Therefore, we adopt PDB as a practical and effective representation.
>
> **On the Theoretical Side**: The entries of the initial $W_0$ are pure noise, and $EW_0^TW_0=\sigma_0^2I_p$. The eigenvalues of $W_0^TW_0$ follow unimodal MP.  As training progresses, $W$ is continuously updated with new gradients, altering the structure of $EW^TW$. Martin \& Mahoney (JMLR, 2021) conducted extensive analyses on the eigenvalue distribution of $W^TW$ during training. They claim that, as training proceeds, eigenvalue distribution becomes bimodal (Figures 11,18 in their paper). We provide a theoretical explanation (PDB for $EW^TW$) for this phenomenon and establish a systematic compression framework.
>
> **5.  Additional moments comparison[T2]**
>
> We compare the first 3 theoretical spectral moments of $W$ with the empirical values on additional neural networks.
>
> |           | |   T5-base: RTE     |       |  |     Bert: SCITAIL  |       |
> | :-------: | :----------: | :---: | :---: | :-----------: | :---: | :---: |
> |  Moments  |    $\gamma_1$     | $\gamma_2$ | $\gamma_3$ |     $\gamma_1$     | $\gamma_2$ | $\gamma_3$ |
> |    PUB    |     0.67     |  0.9  | 1.51  |     0.59      | 0.69  | 1.02  |
> |    PDB    |     0.77     | 1.55  | 4.17  |     0.67      | 1.18  | 2.78  |
> | empirical |     0.72     | 1.83  | 5.35  |     0.71      | 1.38  | 3.95  |

---

> > ### Comment · Reviewer_nNCM · 2025-04-09
> >
> > Thank you for the detailed response and the additional experimental results. My main concern was the lack of evaluation on other models such as LLMs and the comparison with SVD-based methods, which was well addressed in the authors’ reply. For this reason, I have raised my score from 2 to 3.

---

> > > ### Author Response · Authors · 2025-04-09
> > >
> > > We sincerely appreciate your valuable comments and the time you dedicated to reviewing our work. We are delighted that our response and additional experiments have addressed your concerns. Thank you for recognizing our revisions and for your kind support.

---

### Decision · Program_Chairs · 2025-05-01

**Decision:**

Accept (poster)

**Comment:**

This paper proposes a population double bulk (PDB) model for compressing weight matrices in neural networks, improving upon prior approaches such as the population unit bulk (PUB) model.
The authors provide both a theoretical analysis (with random matrix theory) and algorithmic procedures for estimating the parameters of the PDB model.
Experimental results on neural networks demonstrate improved accuracy when using the proposed PDB model compared to the PUB baseline.

The authors have done an excellent job during the rebuttal phase, and all reviewers are now convinced of the significance and merit of the work.

As a result, I recommend acceptance of this paper to ICML 2025.
Please ensure that the additional experiments and clarifications provided during the rebuttal are included in the final version of the paper.